# Does ESG Performance Enhance Financial Flexibility? Evidence from China

**Dingzu Zhang**  **and Luqi Liu \***

School of Economics and Management, Changsha University of Science and Technology, Changsha 410004, China
* Correspondence: custzdz@csust.edu.cn

**Abstract:** Environmental, social, and governance (ESG) performance may be one of the strategies firms adopt to enhance their financial flexibility in response to an increasingly uncertain environment and difficult sustainability conditions. We use A-share listed firms in China from 2015 to 2020 as samples to test the influencing mechanism of ESG performance on financial flexibility. The empirical results indicate that ESG performance significantly enhances financial flexibility. The mechanism results show that financing constraints mediate ESG performance and firms' financial flexibility. The additional analysis suggests that environmental uncertainty and market attention have significant positive moderating effects. That is, the promotion effect of firms in high uncertainty environments is more apparent, and the same is true in high market attention. This study supports instrumental stakeholder theory, signaling, and social impact hypothesis. It has enlightenment significance for firms, investors, and creditors to evaluate ESG performance and government departments to formulate relevant policies.

**Keywords:** ESG performance; financial flexibility; financing constraints; environmental uncertainty; market attention



## 1. Introduction

In today's world, multiple factors such as COVID-19, conflicts, trade frictions, and economic downturn are intertwined. The business environment is significantly more uncertain, and the sustainable development of firms faces severe challenges. Firms should enhance their financial flexibility to cope with increasingly uncertain environments, prevent adverse impacts, and realize sustainable development. Thus, while preventing risks, firms should keep their financial decisions forward-facing and flexible, accurately identify and nimbly grasp the fleeting growth opportunities in uncertain environments, and adjust their business strategies.

Financial flexibility is a systematic, comprehensive ability to actively adapt to environmental changes, deal with system uncertainties, integrate financial resources, and optimize financial behavior decisions [1,2]. When encountering major adverse shocks, firms with sufficient financial flexibility show three advantages: (1) They can invoke and raise funds at a low cost to quickly adjust to the capital structure and avoid financial distress [3]; (2) reduce the negative impact of environmental uncertainty, adapt to the external dynamic environment, improve innovation efficiency, and enhance core competitive advantages [4]; and (3) reserve sufficient resources and capabilities, enhance development potential, proactively create conditions, seize development opportunities, and achieve innovative economic development [5]. Therefore, firms with financial flexibility are more able to cope with risks in the uncertain environment and achieve sustainable development. In other words, firms that can adapt to the adverse environment and operate stably are truly firms with financial flexibility, which is reflected in the small fluctuations of stock returns in the capital market. Adequate financial resources are a necessary condition for firms to cope with environmental uncertainties. Gamba and Triantis [6] believe that financial flexibility can be reserved by

increasing internal cash reserves, enhancing debt financing ability, and improving equity financing ability.

Environmental, social, and governance (ESG) performance is an investment concept and evaluation tool focusing on environmental and social responsibility and corporate governance. It comprehensively assesses a firm's environmental, social responsibility, and corporate governance performances. It provides stakeholders with additional non-financial information, enabling them to better assess the investment risks and benefits and more clearly judge the firm's investment value [7]. According to MSCI's 2021 Global Institutional Investor Survey, global non-ESG equity funds saw cumulative outflows of $700 billion through February 2021, in contrast to ESG equity funds, which saw cumulative inflows of $450 billion. The ESG investment themes and strategies have become the main drivers of global equity inflows, and firms with good ESG performance have become the leading destinations for inflows. It shows that the ESG performance of a firm has become essential information to attract market attention and even change investors' investment strategies. Once good ESG performance information of a firm is captured, interpreted, and evaluated by the market, its value may be discovered and invested in by more creditors or investors. Thus, good ESG performance information brings capital inflow to the firm, which increases the internal cash reserves and financing ability of firms and thereby enhances the financial flexibility of firms. Therefore, there is some correlation between ESG performance and financial flexibility.

Previous studies have found that there is no consistent relationship between ESG performance and corporate earnings realization. Sun and Hou [8] and Engelhardt et al. [9] believe that most emerging market countries have serious problems such as resource shortage, environmental pollution, insufficient regulation and governance, which lead to high ESG risk. Therefore, when making investment decisions in emerging markets, incorporating ESG factors into investment decisions can significantly improve investment performance. On the contrary, most developed market countries have relatively perfect institutions, complete ESG investment systems, and low ESG risks. Therefore, when making investment decisions in developed markets, both ESG investment and non-ESG investment have good performance, and ESG investment has no obvious advantage. Investing in ESG incurs additional costs, and the redistribution of resources from investors to stakeholders violates the classical profit maximization theory, which can harm firm's profitability and market value [10,11]. Management may invest in ESG activities to build personal image at the expense of shareholders, which will exacerbate agency conflicts and damage firm's market value [12,13]. George et al. and Waddock and Graves [14] believe that a firm's reputation is closely related to its social rating, and adopting ESG can provide costs and benefits similar to advertising campaigns. Therefore, strengthening ESG investment can reduce financing costs and increase firm's value and market valuation [15]. Thus, the relationship between ESG performance and financial flexibility is not clear.

China is the second largest economy in the world and an important emerging market. Since China's reform and opening-up, China's economic growth has made an increasing contribution to world economic growth. According to China's National Bureau of Statistics, China's economic growth contributed nearly 30 percent to world economic growth in 2018. China is the biggest contributor to world economic growth. Therefore, the study of China's economy has an important impact on world economic growth. As the micro subject of market, the sustainable development of firms is of great significance to economic growth. Due to the imperfect capital market in China and the widespread problem of information asymmetry, Chinese firms are faced with large financing constraints [16,17]. In the environment of increasing uncertainty, financing constraints lead to the obstruction of external financing of firms, so that firms cannot obtain enough funds to improve financial flexibility to cope with adverse impacts, and sustainable development faces serious challenges.

Given the above problems, based on instrumental stakeholder theory, signaling, and social impact hypothesis, this paper uses A-share listed firms in China from 2015 to 2020 as samples to conduct theoretical analyses and empirical tests on the relationship between

ESG performance and financial flexibility. It is found that ESG performance establishes a close relationship with stakeholders by transmitting positive signals to the outside world, improving organizational legitimacy, and improving operation and management efficiency, positively impacting financial flexibility. Financing constraints are derived from information asymmetry in the incomplete market, which reflects the financing ability of firms. Good ESG performance reduces the information asymmetry to a certain extent, which is conducive to alleviating financing constraints, improving firms' financing ability, and enhancing financial flexibility. Further analysis shows that environmental uncertainty positively moderates the relationship between ESG performance and financial flexibility. Good ESG shows a higher value when the degree of uncertainty is high. As a kind of insurance, it can offset the negative impact of adverse shocks and maintain the competitive advantage of firms, enhancing financial flexibility. Market attention also has a positive moderating effect on them. As a limited resource, market attention can improve the transmission efficiency of ESG information in the capital market, enhance the convincing power of ESG performance, and improve the market reaction to ESG performance. Therefore, in the case of high market attention, the enhancement effect of ESG performance on financial flexibility is more significant.

We contribute to the existing literature in the following areas: (1) we add to the literature by exploring the economic consequences of ESG performance. Existing literature pays more attention to the influence of ESG on financial performance, a firm's value, and stock value, while few studies comprehensively evaluate the relationship and mechanism between ESG performance and financial flexibility. Hang et al. [18] and Xie et al. [19] believe that a firm's environmental responsibility is the performance of meeting stakeholders' expectations. By meeting these needs, firms can obtain financial advantages and improve profitability to improve long-term financial performance. Qureshi et al. [20] believe that the economic and social goals of firms are essentially the same, and they can enhance the loyalty of some stakeholders by undertaking social responsibilities and disclosing ESG information, thus improving firm value. Investors believe that firms with good ESG performance have stronger risk management ability and often give higher appraisal value to such firms, thus increasing the stock market valuation [21]. Based on instrumental stakeholder theory, signaling, and social impact hypothesis, we explore the relationship between ESG performance and financial flexibility and expand the relevant literature on the economic consequences of ESG performance. (2) The present paper contributes to the literature by analyzing the influencing mechanism of financial flexibility. To cope with the increasing environmental uncertainties, firms must reserve financial flexibility to reduce the adverse impact of external shocks, avoid falling into financial distress, and provide funds at any time when favorable investment opportunities appear to seize potential development opportunities [22]. We construct a theoretical analysis framework of ESG performance—financing constraints—financial flexibility; clarify the internal mechanism of ESG performance to improve financial flexibility; and enrich the relevant literature on the influencing factors of financial flexibility. (3) We analyze the differences in the impact of ESG performance on financial flexibility under environmental uncertainty, which provides empirical evidence and theoretical support for financial flexibility decision-making. (4) We also study the differences in the impact of ESG performance on financial flexibility under market attention, providing empirical evidence and theoretical support for firms to choose appropriate stakeholder management strategies.

The rest of the paper is arranged as follows. Section 2 contains an extensive review of the literature and a theoretical framework for the research hypothesis. Section 3 describes the sample, variable measurement, and statistical model. Section 4 shows the results and discusses these results. Section 5 further studies the moderating effects of environmental uncertainty and market concerns on the relationship between ESG performance and financial flexibility. Section 6 is the conclusion and discussion section, which points out the theoretical contribution, practical significance, and future research direction of the study.

## 2. Literature Review and Hypotheses Development

### 2.1. ESG Performance and Financial Flexibility

Financial flexibility is the ability to cope with an uncertain environment, which helps firms to reserve enough funds to cope with possible financial difficulties and investment opportunities. Investment opportunities are favorable and reflect a firm's response to environmental uncertainty and utilization degree [23]. Financial flexibility allows firms to maintain enough spare borrowing capacity, reduce the negative impact of liquidity shocks on investment, avoid financial distress, and stabilize operation [3]. Managers can keep appropriate financial flexibility through financing, leverage and cash holding decisions, that is, to improve the equity financing ability, borrowing ability and cash holding of firms [24].

Based on the theory of sustainable development, ESG comprehensively evaluates the performance of firms in three aspects: environment, social responsibility, and corporate governance, which reflects the sustainable development ability of firms. The instrumental stakeholder theory argues that firms can improve their ESG performance, such as by actively protecting the environment, actively taking social responsibilities, promptly improving governance defects, taking into account the rights and interests of stakeholders, and establishing close relationships with them, to obtain the scarce resources controlled by stakeholders, improve their competitive advantages, and strengthen their profitability [25,26]. These measures further improve their cash flexibility, debt, and equity financing flexibility, thus helping firms to cope with the negative impact of the uncertain environment and find development opportunities in the changing environment.

First, from the signaling perspective, good ESG performance often conveys a signal of sustainable development, which is an important basis for stakeholders to judge firm's operational uncertainty and assess future profitability, cash flow, and credit risk [27]. ESG information can help alleviate information asymmetry, reduce risk expectations of stakeholders, improve credit availability of financial institutions, reduce debt financing costs, and enhance debt financing ability [28]. It also conveys a responsible and ethical signal to the outside world, which is conducive to establishing a good image of the firm. In this way, customers' subjective psychology can be positively improved, their evaluation and satisfaction with products can be enhanced, their purchase intention can be maintained [29], and firm's profitability can be improved. At the same time, it also strengthens the firm brand effect, increases the discrimination with similar firms and products, enhances the competition barrier, improves the profit space, thus increasing free cash flow, which is conducive to enhancing cash flexibility.

Second, from the principal-agent perspective, the ESG concept requires firms to protect shareholders' interests and attach importance to long-term development. Firms with good ESG performance can protect shareholders' rights and interests, which is conducive to alleviating agency conflicts and improves the relationship between management and shareholders, as well as improves the agility of management decision-making [30]. Such firms also tend to have a good internal governance system, which is conducive to restraining major shareholders from encroaching on interests, reducing investors' risks and returns required by investors, thereby helping to improve the equity financing ability of firms [31].

Finally, from the organizational legitimacy perspective, firms can improve their ESG performance and organizational legitimacy by taking on more environmental responsibilities, providing more jobs, and disclosing more ESG-related information, which is conducive to gaining the trust and positive evaluation of the government and establishing a good relationship between governments and firms. In this way, it conveys that the firm has political advantages for the creditors, enhances the creditors' trust, obtains more financing convenience, optimizes the debt maturity structure, and enhances the debt financing ability [32,33]. It can also improve the recognition and confidence of investors, establish a good investor relationship, reduce the cost of equity financing of firms, and enhance equity financing ability [34].

According to the above analysis, having good ESG performance is conducive to establishing close stakeholder relationships, optimizing corporate decision-making, and

enhancing profitability, cash holdings, debt and equity financing flexibility. In this way, we can improve financial flexibility and flexibly cope with the impact of uncertain environment. Based on this information, this paper proposes the following hypothesis:

**Hypotheses 1 (H1).** *Good ESG performance is beneficial to improving financial flexibility of firms.*

### 2.2. The Mediating Effect of Financing Constraints

Financing constraints refer to the difficulty of raising funds relative to investment opportunities, which is rooted in information asymmetry in the incomplete market [35,36]. Studies have found that financing constraints affect business performance, reduce total factor productivity [37], restrict outbound investment, and inhibit R&D and innovation [38], thus damaging the sustainable development ability of firms. With the deepening of environmental uncertainty, Chinese firms are increasingly faced with financing constraints. Financing constraints restrict firms' access to external funds, impair financing ability, and thus are not conducive to improving financial flexibility. It also inhibits the growth of firms and reduces their risk-coping ability [39,40]. It is of great significance for firms to explore how to alleviate financing constraints to improve financial flexibility, enhance risk coping ability and realize sustainable development under uncertain environment.

Given the formation of financing constraints and the role of ESG performance in mitigating information asymmetry, financing constraints may have some mechanism of action in the interaction between ESG performance and financial flexibility. The suitable performance of ESG is beneficial for alleviating information asymmetry in the incomplete market and conveying positive signals of sustainable development to the outside world, enhancing investor confidence, and reducing the rate of return required by investors, thereby reducing financing costs and easing financing constraints [41,42]. Low financing constraints mean that firms can raise enough funds at a reasonable cost, which helps improve their financial flexibility [6].

According to the social impact hypothesis, a firm's behavior that damages social interests, such as environmental pollution, will lead stakeholders to doubt the firm's performance ability and sustainable development ability [43]. To safeguard their interests, the stakeholders who have the implicit claim on firm resources will transform the implicit contract into an explicit contract with higher cost for the firm by increasing the rate of return on investment and liquidated damages [44,45], thus raising the financing cost of the firm and aggravating the financing constraints. On the contrary, firms with good ESG performance can attract and retain more high-quality employees, have higher production and operation efficiency, and have more substantial market competitiveness [46]. Therefore, stakeholders tend to believe that such firms are more powerful and moral, and their operational and default risks are relatively low [47], to proactively reduce expected risks and expected investment returns, which helps firms reduce financing costs, improve financing availability, and alleviate financing constraints.

First, easing financing constraints enhances firms' financing ability, which increases firms' cash stock, broadens their financing channels, improves the timeliness of capital acquisition, and improves financial flexibility. The low degree of a firm's financing constraints means that investors, creditors, and other external stakeholders have a high degree of recognition and trust for firms. They are optimistic about the development prospects of firms, have low investment risk, and are more willing to invest funds in such firms. Therefore, firms with low financing constraints can obtain sufficient capital support at a low cost and then improve the flexibility of debt and equity financing. Second, the easing of financing constraints also brings indirect effects to firms. That is, when the financing ability of firms is enhanced, the capital dilemma is alleviated. The firms have the conditions to increase the R&D investment, expand the production scale, and further expand the market, which enhances the competitiveness of firms and increases the net operating cash flow of firms [48,49]. The improvement of external financing ability and the increase of cash holdings make firms have enough financial resources to buffer the negative impact of envi-

ronmental changes, and meet the investment needs of firms in the future, so as to enhance the ability of firms to cope with uncertain environment and improve financial flexibility.

According to the above analysis, financing constraints will affect financial flexibility through the influence of cash flexibility, debt, and equity financing flexibility. Accordingly, the following hypothesis is put forward:

**Hypotheses 2 (H2).** *Financing constraints play a mediating effect in the process of ESG performance affecting the financial flexibility of firms.*

## 3. Research Design

### 3.1. Sample Selection and Data Source

A-share listed firms in China from 2015 to 2020 were selected as samples to explore the specific path of ESG performance affecting financial flexibility. The ESG rating data are obtained from the WIND database, financial flexibility data come from the RESSET database, and other variable data are from the CSMAR database.

Drawing on the practice of existing research, the following criteria are employed to screen and process the samples: (1) listed financial firms are eliminated; (2) excluded firms with abnormal financial conditions and special treatment by China Securities Regulatory Commission; (3) the samples with missing variable data are removed; and (4) all continuous variables adopted were reduced by 1% to eliminate the influence of outliers. Finally, a total of 11,831 unbalanced panel data of 2859 listed firms were obtained.

### 3.2. Variable Settings

Financial flexibility (FF): Based on the perspective of economic consequences of financial flexibility and Ortiz-de-Mandojan and Bansal [50], this paper adopts the standard deviation of the monthly stock return rate to measure the ability of firms to cope with environmental uncertainties, that is, financial flexibility (FF_STO). First, we download the monthly stock returns of Chinese firms from 2015 to 2020 from RESSET database. Second, EXCEL is used to collate and analyze the data, and samples with missing key data were deleted. Finally, STDEVP function is used to calculate the standard deviation of monthly stock returns in each year. The smaller the standard deviation of the monthly stock return rate, the smaller the financial volatility of firms, the stronger the ability to cope with exogenous shocks, and therefore the higher their financial flexibility. To reduce the bias of the data measurement and based on the perspective of the source of financial flexibility, we also use the sum of excess cash holdings and unused debt financing capacity to measure financial flexibility (FF_C&D). First, we download the cash ratio and asset-liability ratio of Chinese firms from 2015 to 2020 from the CSMAR database. Second, EXCEL is used to calculate the industry average cash holding ratio and industry average debt ratio. Finally, we calculate the financial flexibility (FF_C&D) according to the following formula:

$$FF\_C\&D = (\text{cash holding ratio of firm} - \text{average cash holding ratio of industry}) + MAX (0, \text{average debt ratio of industry} - \text{debt ratio of firm})$$

ESG performance (ESG): Referring to the research of Wang et al. [51], ESG rating data disclosed by Shanghai Huazheng Index Information Service Co. Ltd. is utilized. We download the quarterly ESG rating data of Chinese firms from the WIND database from 2015 to 2020, the 9 grades of this index "C, CC, CCC, B, BB, BBB, A, AA, AAA" were assigned 1–9 respectively, and the natural logarithm was used to measure the quarterly ESG performance. Then, we use quarterly ESG rating data to calculate annual average ESG performance as a measure of corporate ESG performance. Concerning the international mainstream ESG evaluation system and China's national conditions, the ESG index of Huazheng eliminates the indicators that are not applicable or have missing data. It adds the indicators with Chinese characteristics, such as poverty alleviation and China Securities Regulatory Commission punishment, taking the applicability of each indicator into full consideration. The index system has nine levels, including 14 themes, 26 key

indicators and more than 130 underlying data indicators, covering all A-share listed firms in China. The index system has a total of nine grades, from C to AAA. Specifically, it consists of 3 first-level indicators, 14 s-level indicators, and 26 third-level indicators. The first level indicators include three dimensions of environment, society, and corporate governance. The secondary indicators of the environmental dimension include environmental management system, green business objectives, green products, external environmental certification, and environmental violations. The three-level indicators of the environmental dimension include environmental management system, low-carbon plan or target, green business plan, carbon footprint, sustainable products or services, products or firms obtaining environmental certification, and environmental violations. The secondary indicators of the social dimension are institutional system, health and safety, social contribution, and quality management. The three-level indicators of the social dimension are the quality of social responsibility reporting, the goal or plan to reduce safety accidents, negative business events, business accident occurrence trend, social responsibility related donations, employee growth rate, rural revitalization, and product or firm obtaining quality certification. The secondary indicators of corporate governance dimension include system construction, governance structure, business activities, business risks and external disposition. The three-level indicators of corporate governance dimension include corporate self-supervision, affiliated transactions, board independence, tax transparency, asset quality, overall financial credibility, short-term debt repayment risk, equity pledge risk, information disclosure quality, violation events of listed companies and subsidiaries, and violation events of executives and shareholders. For a clearer rating system, please contact Huazheng (https://www.chindices.com/esg-ratings.html#esg-ratings-methodology (accessed on 1 September 2022)).

Financing constraints (FC): According to the research of Kaplan and Zingales [52], the KZ index is used to measure financing constraints. The larger the KZ index, the greater the financing constraints.

Referring to the research of Hoberg et al. [53] and Fahlenbrach et al. [54], 13 variables, such as firm growth, return on equity, capital expenditure, book-to-market ratio, firm risk, internal control, and firm size, are selected as control variables. It covers several aspects, such as the firm's financial condition, operating condition, and governance level, and controls for industry and year to minimize empirical bias. The measurements of the variables are presented in Table 1.

### 3.3. Methodological Remarks

To verify the relationship between ESG performance and financial flexibility, this paper adopts the two-way fixed effects model of controlling the year and industry to test and establish Equation (1):

$$FF_{i,t} = \beta_0 + \beta_1 ESG_{i,t} + \beta_2 Controls_{i,t} + \sum IND + \sum YEAR + \varepsilon_{i,t} \tag{1}$$

where *FF* is the financial flexibility of the explained variable, *ESG* is the ESG performance of the explanatory variable, and Controls is each control variable, see Table 1 for the definition of each variable. When ESG performance promotes financial flexibility, $\beta_1$ is significantly negative.

To test the mediating effect of financing constraints on ESG performance and financial flexibility, Equations (2) and (3) were established based on Equation (1) by referring to the mediating effect test method of Baron and Kenny [55]:

$$FC_{i,t} = \beta_0 + \beta_1 ESG_{i,t} + \beta_2 Controls_{i,t} + \sum IND + \sum YEAR + \varepsilon_{i,t} \tag{2}$$

$$FF_{i,t} = \beta_0 + \beta_1 ESG_{i,t} + \beta_2 FC_{i,t} + \beta_3 Controls_{i,t} + \sum IND + \sum YEAR + \varepsilon_{i,t} \tag{3}$$

Within the equations, *FC* is the financing constraints of the mediating variable. Equation (2) focuses on analyzing whether the ESG performance has an impact on financing constraints,

and Equation (3) is used to explore whether financing constraints play a mediating role. In general, whether ESG performance will have an impact on financial flexibility through financing constraints depends on the significance of regression coefficients of Equations (1)–(3).

**Table 1.** Variable symbols and definitions.

| Symbol | Variable | Variable Definition |
|---|---|---|
| FF_STO | Financial flexibility | Standard deviation of monthly stock returns |
| FF_C&D | Financial flexibility | (cash holding ratio of firm−average cash holding ratio of industry) + MAX (0, average debt ratio of industry − debt ratio of firm) |
| ESG | ESG performance | According to ESG rating "C–AAA" 9 grade ratings are assigned 1–9 and take the natural logarithm |
| FC | Financing constraints | Indicated by the KZ index |
| GROWTH | Firm growth | Sustainable growth rate |
| ROE | Return on equity | Net profit/average net asset |
| CAPEX | Capital spending | The ratio of capital expenditure to total assets |
| BM | Book-to-market ratio | The ratio of a firm's market value to its book value |
| RISK | Firm risk | Operational risk × Financial risk |
| IN | Internal control | Internal control index from the DIB database |
| SIZE | Firm size | The annual asset size of the firm is taken as the natural logarithm |
| SOE | Property rights | If the listed firm is a state-owned firm, the value is 1; otherwise, the value is 0 |
| SHB | Balance of ownership | Shareholding ratio of the 2nd to 5th largest shareholder/Shareholding ratio of the 1st largest shareholder |
| DUAL | Chairman and General Manager | If the chairman and the general manager are the same person, the value is 1; otherwise, the value is 0 |
| OP | Audit opinion | If the audit opinion is the standard audit opinion, the value is 1; otherwise, the value is 0 |
| IBM | Proportion of independent directors | The proportion of independent directors to the total number of board members |
| AGE | The number of years the firm has been listed | The difference between the current year and the year of firm IPO |
| YEAR | Year-fixed effect | Year dummies |
| IND | Industry-fixed effect | Industrial dummies |

## 4. Results

### 4.1. Descriptive Statistics

Table 2 lists the results of descriptive statistical analysis of financial flexibility (FF_STO, FF_C&D), ESG performance (ESG), and other variables. Among them, the mean and median of FF_STO were 0.127 and 0.113, the mean and median of FF_C&D were −0.048 and −0.346, which were close to each other, indicating that the financial flexibility variables of the full sample were approximately normally distributed. The maximum and minimum values of FF_STO are 0.362 and 0.042, and the maximum and minimum values of FF_C&D are −1.155 and 4.939, indicating that the financial flexibility of different firms was significantly different, and the sample discrimination was ideal. There was no noticeable abnormal distribution of the explained variables. The average value of the ESG performance is 1.862, the minimum value is 1.386, the maximum value is 2.197, and the standard deviation is 0.174, indicating that the ESG performance of different listed firms is quite different. The minimum value of financing constraints (FC) is −6.430, the maximum value is 4.233, and the standard deviation is 2.128, indicating that some listed firms face significant financing constraints.

**Table 2.** Descriptive statistics.

| Variable | Observations | Average | S.D | Min | Med | Max |
|----------|-------------|---------|-----|-----|-----|-----|
| FF_STO | 11,831 | 0.127 | 0.062 | 0.042 | 0.113 | 0.362 |
| FF_C&D | 11,831 | −0.048 | 0.988 | −1.155 | −0.346 | 4.939 |
| ESG | 11,831 | 1.862 | 0.174 | 1.386 | 1.792 | 2.197 |
| FC | 11,831 | 0.015 | 2.128 | −6.430 | 0.342 | 4.233 |
| GROWTH | 11,831 | 0.068 | 0.061 | −0.027 | 0.055 | 0.339 |
| ROE | 11,831 | 0.087 | 0.062 | 0.004 | 0.076 | 0.315 |
| CAPEX | 11,831 | 0.047 | 0.043 | 0.001 | 0.034 | 0.211 |
| BM | 11,831 | 0.608 | 0.254 | 0.118 | 0.597 | 1.177 |
| RISK | 11,831 | 2.283 | 2.433 | 0.937 | 1.500 | 17.588 |
| IN | 11,831 | 649.938 | 104.476 | 0 | 667.130 | 810.880 |
| SIZE | 11,831 | 22.337 | 1.307 | 20.148 | 22.142 | 26.438 |
| SOE | 11,831 | 0.312 | 0.463 | 0 | 0 | 1 |
| SHB | 11,831 | 0.784 | 0.636 | 0.010 | 0.617 | 4 |
| DUAL | 11,831 | 0.294 | 0.456 | 0 | 0 | 1 |
| OP | 11,831 | 0.986 | 0.116 | 0 | 1 | 1 |
| IBM | 11,831 | 0.378 | 0.054 | 0.333 | 0.364 | 0.571 |
| AGE | 11,831 | 9.971 | 7.469 | 1 | 8 | 26 |

*4.2. Regression Result Analysis*

4.2.1. ESG Performance and Financial Flexibility

Table 3 shows the regression results of Equations (1)–(3). Columns (1) and (4) show the regression results of Equation (1), the relationship between ESG performance and financial flexibility. Column (2) shows the results of Equation (2), the regression of ESG performance and financing constraints. Columns (3) and (5) show the regression results of Equation (3), ESG performance and financial flexibility, controlling for financing constraints. The regression coefficient of ESG in column (1) is −0.023 and significant at the 1% level. Meanwhile, the regression coefficient of ESG in column (4) is 0.337 and significant at the 1% level, indicating that ESG performance significantly improves financial flexibility, H1 is verified.

4.2.2. The Mediating Effect of Financing Constraints

To test the mediating effect of financing constraints, Baron and Kenny [55] first tested the influence of ESG performance on financing constraints and then further tested the influence of ESG performance on financial flexibility under the control of financing constraints. Finally, the regression coefficients of the explanatory and mediating variables were used for judgment.

Since ESG performance is significantly correlated with financial flexibility in the benchmark regression shown in Columns (1) and (4) of Table 3, it is only necessary to observe the regression coefficient between ESG performance and financing constraints in column (2) and the regression coefficient between ESG performance and financing constraints and financial flexibility after controlling for financing constraints in column (3) and (5). According to the results shown in column (2), the regression coefficient of ESG is −0.817 and is significant at the 1% level, indicating that firms improve their ESG performance to alleviate financing constraints. The regression results of column (3) show that the regression coefficients of ESG and FC are significant at the 1% level. The regression results of column (5) show that the regression coefficient of ESG is significant at the level of 5%, and the regression coefficient of FC is significant at the level of 1%, indicating

that financing constraints play a partial mediating role in the relationship between ESG performance and financial flexibility, which is verified by H2.

**Table 3.** ESG performance, financing constraints, and financial flexibility.

| | (1) | (2) | (3) | (4) | (5) |
|---|---|---|---|---|---|
| | FF_STO | FC | FF_STO | FF_C&D | FF_C&D |
| ESG | −0.023 *** | −0.817 *** | −0.021 *** | 0.337 *** | 0.106 ** |
| | (−7.97) | (−7.92) | (−7.29) | (5.99) | (2.19) |
| FC | | | 0.002 *** | | −0.283 *** |
| | | | (9.47) | | (−66.06) |
| GROWTH | 0.173 *** | 23.045 *** | 0.116 *** | −3.190 *** | 3.335 *** |
| | (12.04) | (45.17) | (7.52) | (−11.48) | (12.96) |
| ROE | −0.174 *** | −31.749 *** | −0.096 *** | 2.324 *** | −6.665 *** |
| | (−11.44) | (−58.76) | (−5.60) | (7.89) | (−23.31) |
| CAPEX | 0.022 ** | 1.070 *** | 0.020 * | −1.868 *** | −1.565 *** |
| | (2.05) | (2.77) | (1.82) | (−8.86) | (−8.68) |
| BM | −0.067 *** | −0.879 *** | −0.065 *** | −0.622 *** | −0.871 *** |
| | (−25.68) | (−9.50) | (−24.86) | (−12.35) | (−20.15) |
| RISK | 0.001 *** | 0.130 *** | 0.001 *** | −0.058 *** | −0.021 *** |
| | (5.31) | (17.99) | (3.71) | (−14.74) | (−6.23) |
| IN | −0.000 | −0.000 | −0.000 | −0.000 ** | −0.000 *** |
| | (−1.01) | (−0.37) | (−0.98) | (−2.49) | (−3.14) |
| SIZE | −0.000 | 0.272 *** | −0.001 | −0.154 *** | −0.077 *** |
| | (−0.29) | (14.58) | (−1.55) | (−15.16) | (−8.79) |
| SOE | −0.002 | 0.128 *** | −0.002 | 0.052 ** | 0.088 *** |
| | (−1.24) | (2.92) | (−1.50) | (2.19) | (4.34) |
| SHB | 0.002 ** | −0.007 | 0.002 ** | 0.024 * | 0.021 * |
| | (2.11) | (−0.28) | (2.14) | (1.70) | (1.82) |
| DUAL | 0.001 | −0.040 | 0.001 | 0.061 *** | 0.050 *** |
| | (1.19) | (−1.10) | (1.29) | (3.07) | (2.93) |
| OP | 0.005 | −0.527 *** | 0.006 | 0.186 ** | 0.037 |
| | (1.15) | (−3.72) | (1.47) | (2.41) | (0.56) |
| IBM | 0.010 | 0.104 | 0.009 | 0.307 * | 0.336 ** |
| | (1.18) | (0.36) | (1.15) | (1.94) | (2.49) |
| AGE | −0.000 *** | 0.007 ** | −0.000 *** | −0.003 ** | −0.001 |
| | (−5.05) | (2.46) | (−5.28) | (−2.27) | (−1.16) |
| _cons | 0.290 *** | −3.274 *** | 0.298 *** | 3.310 *** | 2.383 *** |
| | (24.21) | (−7.68) | (24.91) | (14.26) | (11.99) |
| YEAR/IND | control | control | control | control | control |
| N | 11,831 | 11,831 | 11,831 | 11,831 | 11,831 |
| adj. $R^2$ | 0.409 | 0.375 | 0.413 | 0.139 | 0.372 |

Note: t-statistics in parentheses, * significant at 10% level, ** significant at 5% level, and *** significant at 1% level.

*4.3. Endogeneity Mitigation*

While ESG performance affects financial flexibility, financial flexibility is also likely to affect the ESG investment intensity of firms, thus affecting their ESG performance. In other words, ESG performance and financial flexibility may be mutually causal. The present paper conducts the following tests to solve this problem.

### 4.3.1. The Independent Variables Lag by One Period

We processed the explanatory variable (ESG) with lag phase to obtain L.ESG and substituted it into Equation (1) for regression. The results are shown in column (1) and (2) in Table 4. Through this lead-lag method, the influence of ESG performance of the previous year on financial flexibility of the current year is explored, and the causal relationship between ESG performance and financial flexibility in time is tested, which alleviates the endogeneity problem to a certain extent. The conclusion is consistent with the previous one, indicating that the endogeneity problem does not seriously impact the research of this paper.

**Table 4.** Results of endogeneity test regression analysis.

| | (1) LAG | (2) LAG | (3) PSM | (4) PSM | (5) PSM | (6) PSM | (7) PSM | (8) Heckman | (9) 2SLS |
|---|---|---|---|---|---|---|---|---|---|
| | FF_STO | FF_C&D | FF_STO | FC | FF_STO | FF_C&D | FF_C&D | FF_STO | FF_C&D |
| L.ESG | −0.019 *** (−5.97) | 0.310 *** (4.77) | | | | | | | |
| ESG | | | −0.032 *** (−7.61) | −0.971 *** (−6.53) | −0.030 *** (−7.10) | 0.419 *** (5.10) | 0.143 ** (2.02) | −0.038 *** (−3.82) | 1.453 *** (4.95) |
| FC | | | | | 0.002 *** (5.52) | | −0.284 *** (−42.98) | | |
| GROWTH | 0.158 *** (10.57) | −3.190 *** (−10.21) | 0.193 *** (8.96) | 23.657 *** (30.87) | 0.142 *** (6.07) | −3.138 *** (−7.42) | 3.586 *** (9.08) | 0.210 *** (8.30) | −2.974 *** (−9.99) |
| ROE | −0.146 *** (−9.18) | 2.192 *** (6.58) | −0.176 *** (−7.68) | −32.696 *** (−40.02) | −0.105 *** (−4.03) | 2.364 *** (5.24) | −6.930 *** (−15.64) | −0.286 *** (−10.93) | 2.361 *** (7.67) |
| CAPEX | 0.018 (1.47) | −1.802 *** (−7.21) | 0.016 (0.99) | 1.476 *** (2.58) | 0.013 (0.79) | −2.000 *** (−6.34) | −1.580 *** (−5.84) | −0.007 (−0.40) | −1.787 *** (−8.48) |
| BM | −0.065 *** (−24.59) | −0.696 *** (−12.53) | −0.065 *** (−16.18) | −0.875 *** (−6.13) | −0.063 *** (−15.70) | −0.567 *** (−7.19) | −0.816 *** (−12.02) | −0.095 *** (−24.27) | −0.248 *** (−5.19) |
| RISK | 0.002 *** (6.46) | −0.064 *** (−12.05) | 0.001 *** (3.40) | 0.143 *** (12.40) | 0.001 ** (2.41) | −0.062 *** (−9.78) | −0.022 *** (−3.90) | 0.003 *** (6.84) | −0.053 *** (−12.24) |
| IN | 0.000 (0.89) | −0.000 ** (−2.45) | 0.000 (0.11) | 0.001 * (1.91) | −0.000 (−0.03) | −0.001 *** (−3.43) | −0.000 *** (−2.85) | 0.000 (0.66) | −0.001 *** (−4.69) |
| SIZE | 0.001 (1.07) | −0.142 *** (−12.39) | −0.001 (−0.77) | 0.264 *** (8.60) | −0.001 (−1.42) | −0.177 *** (−10.44) | −0.102 *** (−6.97) | −0.001 (−0.91) | −0.203 *** (−12.84) |
| SOE | −0.001 (−0.50) | 0.035 (1.30) | −0.002 (−1.29) | 0.212 *** (3.31) | −0.003 (−1.54) | 0.041 (1.16) | 0.101 *** (3.34) | −0.007 *** (−2.82) | 0.009 (0.29) |
| SHB | 0.000 (0.47) | 0.019 (1.16) | 0.001 (1.20) | −0.016 (−0.43) | 0.001 (1.24) | 0.020 (0.97) | 0.016 (0.88) | 0.001 (0.62) | 0.016 (1.14) |
| DUAL | 0.001 (0.76) | 0.046 ** (2.01) | 0.003 ** (2.07) | −0.035 (−0.64) | 0.003 ** (2.12) | 0.064 ** (2.12) | 0.054 ** (2.09) | 0.003 (1.57) | 0.052 *** (2.58) |
| OP | −0.001 (−0.14) | 0.107 (1.08) | 0.001 (0.07) | −0.156 (−0.55) | 0.001 (0.11) | 0.013 (0.08) | −0.032 (−0.24) | −0.005 (−0.46) | 0.104 (1.26) |
| IBM | −0.006 (−0.67) | 0.340 * (1.89) | −0.002 (−0.20) | 0.180 (0.40) | −0.003 (−0.23) | 0.226 (0.90) | 0.277 (1.29) | 0.014 (1.02) | 0.341 ** (2.10) |
| AGE | −0.000 (−1.38) | −0.001 (−0.59) | −0.000 ** (−2.53) | 0.004 (0.95) | −0.000 *** (−2.61) | −0.003 (−1.16) | −0.002 (−0.79) | −0.000 ** (−2.44) | −0.005 *** (−3.26) |
| _cons | 0.206 *** (15.99) | 3.344 *** (12.41) | 0.322 *** (14.84) | −3.378 *** (−4.36) | 0.330 *** (15.19) | 4.076 *** (9.54) | 3.116 *** (8.49) | 0.302 *** (8.02) | 2.283 *** (7.89) |
| YEAR/IND | control | control | control | control | control | control | control | / | / |
| /mills lambda | | | | | | | | −0.016 *** (−3.46) | |
| N | 8388 | 8388 | 5158 | 5158 | 5158 | 5158 | 5158 | 11,831 | 11,831 |
| adj. R$^2$ | 0.253 | 0.141 | 0.417 | 0.388 | 0.421 | 0.144 | 0.371 | / | 0.085 |

Note: t-statistics in parentheses, * significant at 10% level, ** significant at 5% level, and *** significant at 1% level.

### 4.3.2. PSM Regression Analysis

The principle of PSM is to match an individual in the treatment group with at least one individual in the control group who is as similar as possible to the individual in other aspects, so as to eliminate the influence of inter-group differences to a certain extent and alleviate the endogeneity problem caused by observable variables. This paper uses the data

after propensity score matching to re-regression. Specifically, the treatment and control groups were divided according to the average value of ESG performance. The control variables mentioned above were selected as matching variables, and the individuals in the control group matched with the treatment group were determined by "1:1 caliper match". The matching results show that the standard deviations of all variables after matching are less than 5%, except for the standard deviations of GROWTH, return on equity (ROE), and equity checks and balances (SHB), which are 9.9%, 11.5%, and 5.2%, respectively. The *t* value corresponding to the average treatment effect (ATT) is −4.23. It is significant at the 1% level, indicating that no systematic difference exists between the treatment group and the control group, which passes the balance test. Equations (1)–(3) were regressed with the matched samples, and the results are shown in columns (3) to (7) in Table 4. The conclusion is consistent with the above analysis.

### 4.3.3. Heckman Two-Stage Regression Analysis

Heckman two-stage regression analysis can deal with the sample selection bias caused by unobserved data, and alleviate the research result bias caused by endogeneity to a certain extent. For this paper, firms with better ESG performance have more perfect management and information disclosure system, and we can obtain enough data for research. On the contrary, the lack of information disclosure system for firms with poor ESG performance makes it impossible to obtain sufficient research data. Therefore, we can only choose to abandon these samples, which leads to the problem of sample selection bias. In this paper, the Heckman two-step method is used for empirical test to alleviate the endogeneity problem caused by sample selection bias. This paper adopts the Heckman two-step method for empirical testing to deal with the endogeneity problem caused by sample selection bias. In the first step, the industry average of ESG performance was used as the instrumental variable, and the probability of the whole sample being rated as good ESG performance was estimated by the Probit model (based on the average ESG rating), and the inverse Mills ratio was calculated. In the second step, the inverse Mills ratio is added to the benchmark regression equation of ESG performance on financial flexibility, namely Equation (1). The results are presented in column (8) of Table 4. The empirical results show that the inverse Mills ratio is significant at the 1% level, indicating that the problem of selective bias exists in the sample and proves the rationality of the Heckman two-stage test. After controlling for sample selection bias, the regression coefficient of ESG is significantly negative at the 1% level, indicating that ESG performance significantly improves financial flexibility. When we used the sum of cash flexibility and debt flexibility to measure financial flexibility (FF_C&D) in Heckman two-stage regression analysis, the inverse Mills ratio was not significant. Therefore, Heckman two-stage method is not applicable. In order to make the conclusions of this paper more reliable, we take the industry average of ESG performance as the instrumental variable and use the two-stage least squares method for regression analysis, as shown in column (9) of Table 4. The results show that the regression coefficient of ESG is significantly positive at the 1% level, which confirms the reliability of the previous conclusion that ESG performance affects financial flexibility.

### *4.4. Robustness Test*
### 4.4.1. Change Measure of Key Variable

The following measurement methods were used to replace the main variables and substituted them into Equation (1) for regression analysis to ensure the robustness of the study.

First, the samples were divided into two groups according to the average value of ESG. The ESG performance was remeasured by assigning 0 below the average value and 1 above the average value. The results are shown in columns (1) and (2) of Table 5, and the regression coefficient of ESG is −0.007 and significant at the 1% level. The conclusion is consistent with the previous one.

**Table 5.** Results of endogeneity test regression analysis.

| | (1) OLS | (2) OLS | (3) OLS | (4) OLS | (5) Tobit |
|---|---|---|---|---|---|
| | FF_STO | FF_C&D | FF_STO | FF_C&D | FF_STO |
| ESG | −0.007 *** | 0.117 *** | −0.016 *** | 0.259 *** | −0.017 *** |
| | (−6.61) | (6.10) | (−6.61) | (3.94) | (−5.08) |
| GROWTH | 0.175 *** | −3.195 *** | 0.175 *** | −3.262 *** | 0.188 *** |
| | (12.15) | (−11.50) | (12.15) | (−11.74) | (10.94) |
| ROE | −0.175 *** | 2.316 *** | −0.175 *** | 2.376 *** | −0.254 *** |
| | (−11.50) | (7.87) | (−11.50) | (8.07) | (−13.98) |
| CAPEX | 0.022 ** | −1.870 *** | 0.022 ** | −1.854 *** | 0.001 |
| | (2.02) | (−8.87) | (2.02) | (−8.79) | (0.10) |
| BM | −0.067 *** | −0.621 *** | −0.067 *** | −0.640 *** | −0.126 *** |
| | (−25.61) | (−12.31) | (−25.61) | (−12.71) | (−42.78) |
| RISK | 0.001 *** | −0.059 *** | 0.001 *** | −0.059 *** | 0.002 *** |
| | (5.54) | (−14.87) | (5.54) | (−15.02) | (6.89) |
| IN | −0.000 * | −0.000 ** | −0.000 * | −0.000 ** | 0.000 |
| | (−1.73) | (−2.07) | (−1.73) | (−2.15) | (0.54) |
| SIZE | −0.000 | −0.154 *** | −0.000 | −0.141 *** | 0.006 *** |
| | (−0.65) | (−15.16) | (−0.65) | (−14.29) | (9.01) |
| SOE | −0.002 | 0.053 ** | −0.002 | 0.068 *** | 0.000 |
| | (−1.53) | (2.23) | (−1.53) | (2.88) | (0.31) |
| SHB | 0.002 ** | 0.023 * | 0.002 ** | 0.023 * | −0.001 |
| | (2.15) | (1.66) | (2.15) | (1.68) | (−1.03) |
| DUAL | 0.001 | 0.061 *** | 0.001 | 0.061 *** | −0.000 |
| | (1.16) | (3.10) | (1.16) | (3.06) | (−0.16) |
| OP | 0.003 | 0.201 *** | 0.003 | 0.191 ** | 0.004 |
| | (0.86) | (2.60) | (0.86) | (2.47) | (0.84) |
| IBM | 0.009 | 0.311 ** | 0.009 | 0.307 * | 0.007 |
| | (1.15) | (1.97) | (1.15) | (1.95) | (0.65) |
| AGE | −0.000 *** | −0.003 ** | −0.000 *** | −0.003 ** | −0.001 *** |
| | (−5.09) | (−2.24) | (−5.09) | (−2.15) | (−6.62) |
| _cons | 0.258 *** | 3.834 *** | 0.269 *** | 3.364 *** | 0.116 *** |
| | (21.29) | (16.34) | (22.67) | (14.42) | (8.77) |
| YEAR/IND | control | control | control | control | control |
| sigma_u | | | | | 0.013 *** |
| | | | | | (12.01) |
| sigma_e | | | | | 0.053 *** |
| | | | | | (130.87) |
| N | 11,831 | 11,831 | 11,831 | 11,831 | 11,831 |
| adj. R$^2$ | 0.408 | 0.139 | 0.408 | 0.138 | / |

Note: t-statistics in parentheses, * significant at 10% level, ** significant at 5% level, and *** significant at 1% level.

Second, according to the risk recommendation of ESG rating, the sample is divided into three groups: the firms rated as "C, CC" as the first group, the firms rated as "CCC, B, BB" as the second group, and the firms rated as "BBB, A, AA, AAA" as the third group. Assign a value of 1 and natural log to the first group, a value of 2 and natural log to the second group, and a value of 3 and natural log to the third group to re-measure ESG performance. The results are presented in columns (3) and (4) of Table 5, and the regression coefficient of ESG is −0.016 and significant at the 1% level. The conclusion is consistent with that mentioned above.

### 4.4.2. Change Regression Model

Considering that financial flexibility measured by the standard deviation of monthly stock returns is left tailed off at 0, the Tobit model is used for re-regression. The Tobit test was conducted on Equation (1), and the results are shown in column (5) of Table 5. The regression coefficient of ESG is −0.017 and significant at the 1% level. The conclusion is consistent with the previous finding, indicating that the empirical results are relatively robust.

## 5. Further Analysis

### 5.1. The Moderating Effect of Environmental Uncertainty

In recent years, with increasing environmental uncertainty, firms' business and default risks have increased [56], and the investment decisions of stakeholders such as investors and creditors are also made with caution. As the external manifestation of the sustainable development ability of firms, ESG performance is increasingly recognized and valued by stakeholders and has become an important factor for stakeholders to consider in their investment decisions.

When environmental uncertainty is high, the investment risk of investors, creditors, and other stakeholders is higher, the investment willingness of risk-averse investors is reduced, and the investment decision of risk-inclined investors is more cautious, which reduces the availability of firm financing. At the same time, high risks require high returns, so the external financing cost of firms increases, financing constraints are aggravated [57], and financial flexibility is lacking. At this time, ESG may show a higher value. The reason is that firms with good ESG performance have a higher organizational reputation, more social capital, and are more trusted by stakeholders, which is equivalent to an insurance investment for firms to get returns when they suffer adverse shocks and offset negative impacts. This can help firms become more competitive [58], thereby reducing financing costs, raising funds, and improving financial flexibility.

However, when environmental uncertainty is low, the investment risk in the capital market is generally low, the stock market volatility is small [59], the investment return is relatively stable, investors have strong investment intentions, and firm financing sources are extensive. At the same time, low risk corresponds to low return, and the external financing cost of firms is low, so the financing constraints are small. At this time, the competitive advantage brought by ESG is not apparent, and the marginal contribution to financial flexibility is small. To summarize, we believe that the higher the degree of environmental uncertainty, the stronger the promotion effect of ESG performance on financial flexibility.

We use the consistency index macroeconomic prosperity index (MACEU) and industry prosperity index (INDEU) in China to measure environmental uncertainty, which will reflect the impact of environmental uncertainty more clearly on the relationship between ESG performance and financial flexibility. The higher the consistency index and industry prosperity index, the lower the environmental uncertainty. We introduce environmental uncertainty variables into Equation (1) for empirical testing, as shown in Equation (4). Manufacturing is the foundation and pillar of national production capacity and national economy, which reflects the development level of social productive forces. It is also the carrier of high-tech industrialization and the primary material basis of people's consumption. The development level of manufacturing firms reflects a country's economic strength. Given the important position of manufacturing firms in the national economy, we further analyze the differences in the role of environmental uncertainty in manufacturing firms when discussing the impact of environmental uncertainty on the relationship between ESG performance and financial flexibility.

Columns (1)~(4) in Table 6 show the regression results of the full sample. The results of columns (1) and (3) show that, the regression coefficient of the independent variable ESG is significantly negative at the 1% level, the regression coefficient of the intersection terms ESG × MACEU is significantly positive at the 1% level, and the regression coefficient of the intersection terms ESG × INDEU is significantly positive at the 5% level. This indicates that environmental uncertainty plays a negative moderating role in the relationship between

ESG performance and financial flexibility. That is, the higher the degree of environmental uncertainty, the stronger the promoting effect of ESG performance on financial flexibility. This conclusion applies to all sample firms. Columns (5)~(8) in Table 6 show the regression results of manufacturing firms. The regression coefficient of the independent variable ESG is significantly negative at the 1% level, the regression coefficient of the intersection terms ESG × MACEU is significantly positive at the 1% level, and the regression coefficient of the intersection terms ESG × INDEU is significantly positive at the 10% level, indicating that environmental uncertainty also plays a negative moderating role in manufacturing firms. That is, the higher the environmental uncertainty, the more helpful ESG performance is for firms to cope with risks and enhance financial flexibility. The results of columns (2) and (6) show that the regression coefficient of the independent variable ESG is significantly positive at the 1% level, and the regression coefficient of the cross-multiplicative term ESG × MACEU is significantly positive at the 10% level, indicating that environmental uncertainty negatively moderates the relationship between ESG performance and financial flexibility. In other words, the higher the environmental uncertainty is, the weaker the enhancing effect of ESG performance on financial flexibility. This may be because when the degree of environmental uncertainty is high, the cash holding and financing ability of firms decreases. At this time, ESG investment further reduces the cash holding and spare debt ability of firms, thus hindering the improvement of financial flexibility. This conclusion is applicable to all sample firms and manufacturing firms.

$$FF_{i,t} = \beta_0 + \beta_1 ESG_{i,t} + \beta_2 MACEU_{i,t}(\beta_2 INDEU_{i,t}) + \beta_3 ESG_{i,t} * MACEU_{i,t} \\ (\beta_3 ESG_{i,t} * INDEU_{i,t}) + \beta_4 Controls_{i,t} + \sum IND + \sum YEAR + \varepsilon_{i,t} \tag{4}$$

### 5.2. The Moderating Effect of Market Attention

Restricted by time and energy, market participants often cannot focus on all the capital market firms. Instead, they can only invest time and energy in the firms that attract their attention the most. In this way, they can fully understand firms' operating conditions and development potential and make decisions accordingly, whereas firms that cannot attract their attention are excluded [60,61]. Therefore, as a limited resource, market attention can improve the transmission efficiency of ESG performance in the capital market, enhance the convincing power of ESG performance, and improve the possibility of investment to strengthen the release effect of ESG performance on financing constraints and improve financial flexibility [62].

Market attention has a megaphone function. As important participants in the capital market, analysts are important information intermediaries [63]. They have the strength and motivation to fully excavate and sort out various financial and non-financial information, including ESG information of the firms concerned, and make a professional interpretation and an objective evaluation of relevant information. Finally, they release research reports to convey relevant information to the capital market to alleviate the degree of information asymmetry and improve the overall operating efficiency of the capital market [64,65]. Therefore, analysts can help improve the transmission efficiency of ESG information of the concerned firm in the capital market, expand the influence scope of ESG information of the firm, and enable it to obtain timely market feedback to strengthen the market reaction to ESG performance of the firm [66], alleviating financing constraints.

Market attention also has an authenticating function. The authenticity of the ESG performance of firms with high market attention is more likely to be recognized by the capital market, which can more effectively convey the value signal of ESG to the outside world, improve the value evaluation of ESG performance by the capital market [66], and further enhance the promotion effect of ESG performance on financial flexibility. Investors pay close attention to each other's investment decisions [67]. Institutional investors with advantages in professional knowledge and information mining are important reference groups for other investors to make decisions. Institutional investors tend to invest in firms with good ESG performance, which affects the investment decisions of other investors [68],

promotes other investors' understanding and recognition of ESG information, and guides them to invest more funds in firms with good ESG performance, strengthen the release effect of ESG performance on financing constraints, and promote the relationship between ESG performance and financial flexibility.

**Table 6.** The moderating effect of environmental uncertainty.

| | Full Sample | | | | Manufacturing | | | |
|---|---|---|---|---|---|---|---|---|
| | **(1) FF_STO** | **(2) FF_C&D** | **(3) FF_STO** | **(4) FF_C&D** | **(5) FF_STO** | **(6) FF_C&D** | **(7) FF_STO** | **(8) FF_C&D** |
| ESG | −0.023 *** | 0.338 *** | −0.023 *** | 0.338 *** | −0.022 *** | 0.358 *** | −0.023 *** | 0.369 *** |
| | (−7.94) | (6.01) | (−8.00) | (6.01) | (−6.30) | (5.29) | (−6.57) | (5.36) |
| MACEU | −0.004 *** | −0.001 | | | −0.004 *** | −0.003 | | |
| | (−42.61) | (−0.57) | | | (−32.37) | (−1.28) | | |
| ESG × MACEU | 0.001 *** | 0.013 * | | | 0.002 *** | 0.015 * | | |
| | (3.33) | (1.86) | | | (3.58) | (1.65) | | |
| INDEU | | | 0.000 | 0.000 | | | −0.024 *** | −0.019 |
| | | | (0.17) | (0.14) | | | (−32.51) | (−1.29) |
| ESG × INDEU | | | 0.001 ** | −0.010 | | | 0.001 * | −0.009 |
| | | | (2.42) | (−1.61) | | | (1.90) | (−0.99) |
| GROWTH | 0.173 *** | −3.188 *** | 0.172 *** | −3.177 *** | 0.174 *** | −2.962 *** | 0.174 *** | −2.952 *** |
| | (12.05) | (−11.47) | (11.97) | (−11.42) | (10.74) | (−9.37) | (10.70) | (−9.33) |
| ROE | −0.174 *** | 2.329 *** | −0.173 *** | 2.312 *** | −0.180 *** | 2.425 *** | −0.180 *** | 2.410 *** |
| | (−11.42) | (7.91) | (−11.39) | (7.85) | (−10.64) | (7.35) | (−10.63) | (7.30) |
| CAPEX | 0.022 ** | −1.872 *** | 0.023 ** | −1.873 *** | 0.020 | −2.078 *** | 0.021 * | −2.078 *** |
| | (2.02) | (−8.88) | (2.08) | (−8.88) | (1.59) | (−8.33) | (1.67) | (−8.33) |
| BM | −0.067 *** | −0.622 *** | −0.067 *** | −0.625 *** | −0.068 *** | −0.701 *** | −0.068 *** | −0.705 *** |
| | (−25.68) | (−12.34) | (−25.59) | (−12.40) | (−22.52) | (−11.88) | (−22.46) | (−11.93) |
| RISK | 0.001 *** | −0.058 *** | 0.001 *** | −0.058 *** | 0.001 *** | −0.055 *** | 0.001 *** | −0.055 *** |
| | (5.33) | (−14.73) | (5.33) | (−14.75) | (4.69) | (−11.94) | (4.68) | (−11.97) |
| IN | −0.000 | −0.000 ** | −0.000 | −0.000 ** | −0.000 | −0.000 | −0.000 | −0.000 |
| | (−1.05) | (−2.51) | (−1.02) | (−2.49) | (−1.29) | (−1.52) | (−1.22) | (−1.46) |
| SIZE | −0.000 | −0.154 *** | −0.000 | −0.154 *** | 0.001 * | −0.172 *** | 0.001 * | −0.171 *** |
| | (−0.33) | (−15.18) | (−0.25) | (−15.18) | (1.83) | (−13.74) | (1.84) | (−13.69) |
| SOE | −0.001 | 0.053 ** | −0.001 | 0.052 ** | 0.000 | 0.063 ** | 0.000 | 0.062 ** |
| | (−1.20) | (2.21) | (−1.22) | (2.18) | (0.32) | (2.17) | (0.24) | (2.14) |
| SHB | 0.002 ** | 0.024 * | 0.002 ** | 0.023 * | 0.001 * | 0.037 ** | 0.001 * | 0.037 ** |
| | (2.12) | (1.71) | (2.13) | (1.68) | (1.66) | (2.24) | (1.72) | (2.22) |
| DUAL | 0.001 | 0.061 *** | 0.001 | 0.061 *** | 0.001 | 0.053 ** | 0.001 | 0.053 ** |
| | (1.20) | (3.08) | (1.19) | (3.08) | (0.71) | (2.30) | (0.66) | (2.28) |
| OP | 0.004 | 0.185 ** | 0.004 | 0.187 ** | 0.009 * | 0.208 ** | 0.009 * | 0.208 ** |
| | (1.12) | (2.39) | (1.12) | (2.42) | (1.86) | (2.23) | (1.93) | (2.23) |
| IBM | 0.010 | 0.308 * | 0.010 | 0.304 * | 0.008 | 0.013 | 0.008 | 0.015 |
| | (1.20) | (1.95) | (1.20) | (1.92) | (0.86) | (0.07) | (0.86) | (0.08) |
| AGE | −0.000 *** | −0.003 ** | −0.000 *** | −0.003 ** | −0.001 *** | −0.004 ** | −0.001 *** | −0.004 ** |
| | (−5.06) | (−2.28) | (−5.08) | (−2.25) | (−6.98) | (−2.14) | (−6.92) | (−2.13) |
| _cons | 0.217 *** | 3.938 *** | 0.247 *** | 3.951 *** | 0.181 *** | 4.140 *** | 0.095 *** | 4.059 *** |
| | (17.68) | (16.59) | (19.78) | (16.34) | (13.08) | (15.34) | (6.85) | (15.05) |
| YEAR/IND | control | control | control | control | control | control | control | control |
| N | 11,831 | 11,831 | 11,831 | 11,831 | 8408 | 8408 | 8408 | 8408 |
| adj. $R^2$ | 0.409 | 0.139 | 0.409 | 0.139 | 0.383 | 0.159 | 0.382 | 0.159 |

Note: t-statistics in parentheses, * significant at 10% level, ** significant at 5% level, and *** significant at 1% level.

To reflect the relationship more clearly between market focus on ESG performance and financial flexibility, we use the number of analysts following (ANATT) and the proportion of institutional investors holding (INSATT) to measure the attention of market participants to firms. The more analysts that follow, or institutional investors hold shares, the more attention market participants pay to companies. We introduce the market concern variable into Equation (1) to test its impact on the relationship between ESG performance and financial flexibility, as shown in Equation (5).

Columns (1)~(4) of Table 7 show the regression results of the entire sample. The results of columns (1) and (3) show that, the regression coefficients of the independent variable ESG

and the intersection terms ESG × ANATT and ESG × INSATT are all significantly negative at the 1% level, indicating that the attention of market participants to firms significantly affects the relationship between ESG performance and financial flexibility. That is, the more attention the market gives, the more the market gives. The enhancement effect of ESG performance on financial flexibility is more robust, and this conclusion applies to all sample firms. Furthermore, we explore whether the positive moderating effect of market attention on ESG performance and financial flexibility is different among manufacturing firms. Columns (5) and (8) in Table 7 show the regression results of manufacturing firms. The results of columns (5) and (7) show that, the regression coefficients of the independent variable ESG and the intersection terms ESG × ANATT and ESG × INSATT are all significantly negative at the 5% level, indicating that, under high market concern, ESG performance has a more apparent enhancing effect on financial flexibility in manufacturing firms.

$$
FF_{i,t} = \beta_0 + \beta_1 ESG_{i,t} + \beta_2 ANATT_{i,t}(\beta_2 INSATT_{i,t}) + \beta_3 ESG_{i,t} * ANATT_{i,t} \\
(\beta_3 ESG_{i,t} * INSATT_{i,t}) + \beta_4 Controls_{i,t} + \sum IND + \sum YEAR + \varepsilon_{i,t}
\tag{5}
$$

**Table 7.** The moderating effect of market attention.

| | Full Sample | | | | Manufacturing | | | |
|---|---|---|---|---|---|---|---|---|
| | (1) FF_STO | (2) FF_C&D | (3) FF_STO | (4) FF_C&D | (5) FF_STO | (6) FF_C&D | (7) FF_STO | (8) FF_C&D |
| ESG | −0.024 *** | 0.344 *** | −0.023 *** | 0.338 *** | −0.023 *** | 0.364 *** | −0.023 *** | 0.357 *** |
| | (−8.15) | (6.08) | (−8.03) | (6.01) | (−6.44) | (5.33) | (−6.48) | (5.26) |
| ANATT | −0.001 | −0.009 | | | −0.001 | −0.011 | | |
| | (−1.10) | (−0.89) | | | (−1.46) | (−0.89) | | |
| ESG × ANATT | −0.007 *** | 0.032 | | | −0.006 ** | 0.023 | | |
| | (−3.48) | (0.77) | | | (−2.17) | (0.44) | | |
| INVH | | | −0.009 *** | 0.053 | | | −0.010 *** | 0.020 |
| | | | (−4.18) | (1.23) | | | (−4.03) | (0.41) |
| ESG × INVH | | | −0.033 *** | 0.160 | | | −0.032 ** | 0.015 |
| | | | (−3.03) | (0.76) | | | (−2.34) | (0.05) |
| GROWTH | 0.171 *** | −3.196 *** | 0.166 *** | −3.151 *** | 0.173 *** | −2.969 *** | 0.167 *** | −2.947 *** |
| | (11.92) | (−11.48) | (11.50) | (−11.28) | (10.64) | (−9.38) | (10.26) | (−9.27) |
| ROE | −0.171 *** | 2.366 *** | −0.168 *** | 2.287 *** | −0.177 *** | 2.468 *** | −0.174 *** | 2.405 *** |
| | (−11.06) | (7.92) | (−10.98) | (7.73) | (−10.28) | (7.37) | (−10.25) | (7.25) |
| CAPEX | 0.023 ** | −1.845 *** | 0.023 ** | −1.872 *** | 0.022 * | −2.044 *** | 0.022 * | −2.076 *** |
| | (2.06) | (−8.71) | (2.11) | (−8.87) | (1.73) | (−8.14) | (1.72) | (−8.31) |
| BM | −0.069 *** | −0.636 *** | −0.069 *** | −0.610 *** | −0.071 *** | −0.721 *** | −0.071 *** | −0.698 *** |
| | (−24.59) | (−11.70) | (−26.12) | (−11.91) | (−21.49) | (−11.26) | (−22.98) | (−11.64) |
| RISK | 0.001 *** | −0.059 *** | 0.001 *** | −0.058 *** | 0.001 *** | −0.055 *** | 0.001 *** | −0.055 *** |
| | (5.27) | (−14.78) | (5.12) | (−14.68) | (4.51) | (−11.99) | (4.49) | (−11.94) |
| IN | −0.000 | −0.000 ** | −0.000 | −0.000 ** | −0.000 | −0.000 | −0.000 | −0.000 |
| | (−1.02) | (−2.46) | (−0.91) | (−2.52) | (−1.18) | (−1.46) | (−1.07) | (−1.48) |
| SIZE | 0.001 | −0.149 *** | 0.001 | −0.160 *** | 0.002 ** | −0.164 *** | 0.002 *** | −0.173 *** |
| | (0.88) | (−11.97) | (1.58) | (−14.54) | (2.57) | (−10.55) | (3.33) | (−12.91) |
| SOE | −0.002 | 0.050 ** | 0.000 | 0.043 * | 0.000 | 0.061 ** | 0.002 | 0.060 ** |
| | (−1.36) | (2.10) | (0.18) | (1.71) | (0.13) | (2.09) | (1.31) | (1.97) |
| SHB | 0.002 ** | 0.024 * | 0.001 | 0.026 * | 0.001 * | 0.037 ** | 0.001 | 0.038 ** |
| | (2.23) | (1.70) | (1.41) | (1.88) | (1.74) | (2.26) | (1.03) | (2.28) |
| DUAL | 0.001 | 0.061 *** | 0.001 | 0.062 *** | 0.001 | 0.053 ** | 0.001 | 0.053 ** |
| | (1.26) | (3.07) | (1.00) | (3.13) | (0.73) | (2.29) | (0.49) | (2.29) |
| OP | 0.004 | 0.193 ** | 0.004 | 0.191 ** | 0.008 * | 0.216 ** | 0.008 * | 0.210 ** |
| | (0.91) | (2.49) | (0.90) | (2.47) | (1.76) | (2.31) | (1.71) | (2.25) |
| IBM | 0.010 | 0.302 * | 0.010 | 0.308 * | 0.008 | 0.010 | 0.008 | 0.017 |
| | (1.23) | (1.91) | (1.17) | (1.95) | (0.87) | (0.05) | (0.85) | (0.09) |
| AGE | −0.000 *** | −0.004 ** | −0.000 *** | −0.003 ** | −0.001 *** | −0.004 ** | −0.001 *** | −0.004 ** |
| | (−5.18) | (−2.36) | (−5.37) | (−2.17) | (−7.03) | (−2.25) | (−7.07) | (−2.11) |
| _cons | 0.233 *** | 3.827 *** | 0.226 *** | 4.057 *** | 0.191 *** | 3.993 *** | 0.186 *** | 4.188 *** |
| | (15.92) | (13.48) | (17.18) | (15.95) | (11.12) | (11.94) | (12.49) | (14.45) |
| YEAR/IND | control | control | control | control | control | control | control | control |

**Table 7.** *Cont.*

| | Full Sample | | | | Manufacturing | | | |
|---|---|---|---|---|---|---|---|---|
| | (1) FF_STO | (2) FF_C&D | (3) FF_STO | (4) FF_C&D | (5) FF_STO | (6) FF_C&D | (7) FF_STO | (8) FF_C&D |
| N | 11,831 | 11,831 | 11,831 | 11,831 | 8408 | 8408 | 8408 | 8408 |
| adj. $R^2$ | 0.409 | 0.139 | 0.410 | 0.139 | 0.382 | 0.159 | 0.383 | 0.159 |

Note: t-statistics in parentheses, * significant at 10% level, ** significant at 5% level, and *** significant at 1% level.

## 6. Discussion

At present, the degree of environmental uncertainty is deepening, and the sustainable development of firms is facing severe challenges. In the uncertain environment, firms need to reserve appropriate financial flexibility to cope with risks, stable operation, seize potential investment opportunities, and achieve sustainable development. ESG performance provides additional information to stakeholders and is an important consideration for stakeholders in their investment decisions. Good ESG performance can help firms gain the trust and support of stakeholders, thereby enhancing the cash holding and external financing ability.

The results in Table 3 support H1, good ESG performance is beneficial to improving financial flexibility. Our results support the views of Hur et al. [29], Olsen [32], and Ng and Rezaee [34], providing evidence of instrumental stakeholder theory and signaling theory. Firms can obtain scarce resources for sustainable development by managing stakeholder relationships. For example, firms with good relations with the government can convey political advantages to the market, gain recognition and trust from creditors and investors, and thus have more financing convenience, lower financing cost, and stronger financing ability. Good ESG performance can convey a positive signal of responsibility and ethics to the outside world, strengthen brand effect, enhance customer satisfaction and purchase intention of products, and improve corporate profitability and cash flow level. The improvement of financing ability, profitability, and cash flow makes firms have more sufficient resources to cope with adverse shocks, so as to improve the ability of firms to cope with environmental uncertainties and enhance financial flexibility.

Information asymmetry is the root cause of financing constraints. Due to the imperfect capital market in China, Chinese firms are generally faced with financing constraints. Under the increasingly severe environmental uncertainty, how to alleviate financing constraints, enhance financing ability, improve financial flexibility, and avoid falling into financial distress is an urgent problem for firms to achieve sustainable development. The results in Table 3 also support H2, financing constraints have a mediating role in the process of ESG performance affecting financial flexibility. Our results support Deng et al. [69] and Latane [44] and provide evidence for social influence theory. Behaviors that violate the ESG concept will make stakeholders doubt the sustainable development ability of the firm, thereby increasing the return required by investors, raising the financing cost of the firm, aggravating financing constraints, and thus hindering the improvement of financial flexibility. On the contrary, improving ESG performance of firms can help improve organizational reputation, reduce information asymmetry, thus enhance investor confidence, alleviate financing constraints, raise funds, and improve financial flexibility. Therefore, ESG performance improves financial flexibility by releasing financing constraints.

Environmental uncertainty increases the risk of firm operation and default, deteriorates the financing environment and operation environment of firms, and poses a great threat to sustainable development. Then, does the degree of environmental uncertainty affect the relationship between ESG performance and financial flexibility? The results in Table 6 suggest that ESG performance is more conducive to improving financial flexibility when the degree of environmental uncertainty is high. However, this enhancement effect is only reflected in improving the risk coping ability of firms, while hindering firms to hold cash and reserve surplus debt capacity. This may be because ESG performance acts as an insurance against the negative effects of an uncertain environment [58]. But firms may

take on more debt in response to external shocks, and investing in ESG further reduces the firm's cash, thereby compromising its ability to hold cash and spare debt.

Market attention has loudspeaker and authentication functions, which can improve the speed and authority of ESG information transmission, thus enhancing the market response of ESG information. As a limited resource, the degree of market attention may have an impact on the relationship between ESG performance and financial flexibility. The results in Table 7 suggest that the impact of ESG performance on financial flexibility is stronger when market concern is high. With the increase of market attention, the transmission efficiency and persuasion of ESG performance in the capital market are improved, which is beneficial to strengthen the release effect of ESG performance on financing constraints and better improve financial flexibility [62].

## 7. Conclusions

Increasingly major environmental problems, the global spread of COVID-19, wars and conflicts, and trade barriers have made the environment rife with uncertainty and has led to firms facing significant challenges when working toward the goal of sustainable development. In this case, firms need to improve financial flexibility to meet future capital needs, avoid financial difficulties, and seize the development opportunities when they arise. Firms with good ESG performance can gain favor from investors, enhance financing ability raise funds, and improve financial flexibility. We explore the relationship between ESG performance and financial flexibility in this case. The paper finds that ESG performance and financial flexibility are significantly positively correlated. The mechanism results show that financing constraints mediate ESG performance and firms' financial flexibility. The additional analysis suggests that environmental uncertainty and market attention have a significant positive moderating effect. The promoting effect is more pronounced when the firm is in a high uncertainty environment, and the same is true with high market attention.

Our results have important implications for corporate managers, policymakers, investors, and creditors. (1) In the case of environmental uncertainty, the firm can actively participate in ESG activities, disclose relevant information, and pay close attention to market participants, especially the institutional investors and analysts, to understand the relevant information. Thus, firms can better alleviate financing constraints, improve financial flexibility, enhance the ability to cope with uncertain environments, and achieve sustainable development. (2) Policymakers can establish relevant regulations and evaluate ESG disclosure to mobilize the initiative of the ESG responsibility of firms. In this way, ESG information can provide effective data support for investors to make decisions, guide capital to an ethical, responsible, and sustainable field, and improve the efficiency of resource allocation. (3) Investors and creditors can incorporate ESG factors into their investment strategies to identify firms with low operational and default risks, reducing investment risk and improving return on investment.

Our article is not without its limitations. First, we only use firms from mainland China as samples to explore the relationship between ESG performance and financial flexibility, so an analysis based on international data is lacking. Second, in terms of the measurement of the dependent variable, we use the methods commonly used in the literature—the standard deviation of monthly stock return and the sum of cash flexibility and debt flexibility to measure financial flexibility indirectly. We fail to use creative methods to measure financial flexibility directly, and cannot provide suggestions for how much financial flexibility to reserve. Third, considering that we only use Chinese firms as samples, we also only use ESG rating data disclosed by Huazheng which is more consistent with China's national conditions, failing to compare the impact of ESG rating data from different institutions on conclusions, and failing to provide suggestions for firms on how to invest in ESG. Fourth, in addition to analyzing all samples as a whole, we only consider the differences in the relationship between ESG performance and the financial flexibility of manufacturing firms under different environmental uncertainties and market concerns without conducting in-depth research on more industries. Future research can use international data, a broader

sample, and make a more detailed division of sample industries to reveal the relationship between ESG performance and financial flexibility. Future research could also take more creative approaches to measure financial flexibility and ESG performance directly and provide a more profound and straightforward analysis of the impact of ESG performance on financial flexibility.

**Author Contributions:** Conceptualization, D.Z. and L.L.; methodology, D.Z. and L.L.; software, D.Z. and L.L.; validation, D.Z. and L.L.; formal analysis, D.Z. and L.L.; investigation, L.L.; resources, D.Z. and L.L.; data curation, D.Z. and L.L.; writing—original draft preparation, L.L.; writing—review and editing, D.Z. and L.L.; visualization, D.Z. and L.L.; supervision, D.Z.; project administration, D.Z.; funding acquisition, D.Z. All authors have read and agreed to the published version of the manuscript.

**Funding:** This research was supported by the National Social Science Foundation of China (grant number: 18BJY023) and Research Center of Intelligent Accounting (grant number: 21ckyb02).

**Institutional Review Board Statement:** Not applicable.

**Informed Consent Statement:** Not applicable.

**Data Availability Statement:** The ESG data come from ESG disclosure scores. The financial data comes from the China Stock Market and Accounting Research (CSMAR) Database, WIND database and RESSET database.

**Conflicts of Interest:** The authors declare no conflict of interest.

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
