# Peer review of "Does ESG Performance Enhance Financial Flexibility? Evidence from China"

_sustainability, doi:10.3390/su141811324_

Round 1

Reviewer 1 Report

Although the research looks methodologically looks, the authors have unilaterally emphasized only positive nature of ESG performance evaluation instead of using a more neutral scientific approach. From an ESG finance perspective, a problem might be that this study is based on a relatively short time period (2015-2020) over which conclusions on financial flexibility have been formulated. Is it possible that in a short-term period, the results might be sensitive to specific macroeconomic conditions?  For example, energy prices may have had a certain tendency to increase or decrease or the economy may have been in a period of growth or decline. In my opinion, the authors should also critically assess the publications related with ESG criticism and provide their arguments in this respect.  Currently, the authors have distanced themselves from critical evaluation and enthusiastically praised only positive aspects. As a result, there is a lot of redundancy and repetition of theory in Introduction and Literature review. Also, the justification for H1 is insufficient and needs to be improved.

Author Response

Dear reviewer:

   Thank you for your kind and good suggestions!

   We have carefully revised it according to your guidance. In this version of the manuscript, we have mainly made the following adjustments:

Point 1: Although the research looks methodologically looks, the authors have unilaterally emphasized only positive nature of ESG performance evaluation instead of using a more neutral scientific approach. From an ESG finance perspective, a problem might be that this study is based on a relatively short time period (2015-2020) over which conclusions on financial flexibility have been formulated. Is it possible that in a short-term period, the results might be sensitive to specific macroeconomic conditions?  For example, energy prices may have had a certain tendency to increase or decrease or the economy may have been in a period of growth or decline. In my opinion, the authors should also critically assess the publications related with ESG criticism and provide their arguments in this respect.  Currently, the authors have distanced themselves from critical evaluation and enthusiastically praised only positive aspects. As a result, there is a lot of redundancy and repetition of theory in Introduction and Literature review.

Response 1: We fully considered the reviewer's valuable comments, read more literature, and deleted duplicates and redundant parts. In the introduction, we critically assess the publications related with ESG criticism and provide their arguments.

Previous studies have found that there is no consistent relationship between improved ESG performance and corporate earnings realization. Sun & Hou (2021) and Engelhardt et al. (2021) believe that most emerging market countries have serious problems such as resource shortage, environmental pollution, insufficient regulation and governance, which lead to high ESG risk. Therefore, when making investment decisions in emerging markets, incorporating ESG factors into investment decisions can significantly improve investment performance. On the contrary, most developed market countries have relatively perfect institutions, complete ESG investment systems and low ESG risks. Both ESG investment and non-ESG investment have good performance, and ESG investment has no obvious advantage. By analyzing relevant literatures, Friede et al. (2015) found that 90% of the papers showed non-negative correlation between ESG and financial performance, and 62.6% of the papers showed positive correlation between ESG and financial performance. Investing in ESG incurs additional costs, and the redistribution of resources from investors to stakeholders violates the classical profit maximization theory, which can harm corporate profitability and market value (Artiach et al.,2010; Friedman,1970). Management may invest in ESG activities to build personal image at the expense of shareholders, which will exacerbate agency conflicts and damage the market valuation of firms (Krüger,2015; Bae et al.,2021). George et al. and Waddock & Graves (IONESCU,2019; Waddock & Graves,1997) believe that a firm's reputation is closely related to its social rating, and adopting ESG can provide costs and benefits similar to advertising campaigns. Therefore, strengthening ESG investment can reduce financing costs, increase firm value and improve market valuation (Oikonomou et al.,2014; Giese et al.,2017). However, it takes some time for the information of ESG performance score to be fully absorbed by market participants and reflected in stock price changes (Usman,2020). Thus, the relationship between ESG performance and financial flexibility is not clear (lines 67-90).

Point 2: The justification for H1 is insufficient and needs to be improved.

  Response 2: Thanks for the reviewer's valuable comments, we have read more literatures, improved our theoretical analysis, and enhanced the rationality of the paper (see Section 2).

We hope that our amendments will satisfy you. Once again, thank you for your efforts in revising and improving the quality of the manuscript.

    Best Regards

         Dingzu Zhang

         Luqi Liu

2022.8.28

Reviewer 2 Report

Review for sustainability-1889605

Does ESG Performance Enhance Financial Flexibility? Evidence from China

The paper under review examines the association between ESG performance and financial flexibility, defined as the flexibility to cope with uncertain environments. It also investigates the role of financial constraints as the mediating factor and two moderating factors (environmental uncertainty and market attention). My comments below are intended to help the authors to further strengthen the inferences of the paper.

1.     The authors argue that “corporate cash holdings, debt financing ability, and equity financing ability are important source channels of financial flexibility” (see page 2). However, it is unclear how standard deviation of stock returns captures the construct of financial flexibility. Note that stock return volatility typically proxies for risk or uncertainty in the literature. I suggest the authors to consider employing alternative and more direct measures of financial flexibility that better capture the underlying construct.

2.     As the authors acknowledge, endogeneity is a concern when interpreting the association between ESG performance and financial flexibility. I suggest the authors to clearly explain how the three tests in Section 4.3 help address the endogeneity concern in their setting. In addition, I suggest the authors to include firm fixed effects to further mitigate the influence of firm-level time-invariant factors.

3.     I suggest the authors to discuss why examining their research question in the Chinese setting provides unique insights beyond studies that focus on the US or other countries.

Author Response

Dear reviewer:

Thank you for your kind and good suggestions!

We have carefully revised it according to your guidance. In this version of the manuscript, we have mainly made the following adjustments:

Point 1: The authors argue that “corporate cash holdings, debt financing ability, and equity financing ability are important source channels of financial flexibility” (see page 2). However, it is unclear how standard deviation of stock returns captures the construct of financial flexibility. Note that stock return volatility typically proxies for risk or uncertainty in the literature. I suggest the authors to consider employing alternative and more direct measures of financial flexibility that better capture the underlying construct.

Response 1: Thanks for the reviewer's valuable comments, we have made the following modifications on the basis of the original. We improved the expression of financial flexibility in the paper and added another measurement method of financial flexibility to make our variable selection more reasonable.

Financial flexibility is a systematic, comprehensive ability to actively adapt to environmental changes, deal with system uncertainties, integrate financial resources, and optimize financial behavior decisions (Golden & Powell,2000; Zhao & Zhang,2010). FF firms are more able to cope with risks in the uncertain environment and achieve sustainable development. In other words, firms that can adapt to the adverse environment and operate stably are truly firms with financial flexibility, which is reflected in the small fluctuations of stock returns in the capital market. Adequate financial resources are a necessary condition for enterprises to cope with environmental uncertainties. Gamba & Triantis (2008) believes that financial flexibility can be reserved by increasing internal cash reserves, enhancing debt financing ability, and improving equity financing ability (lines 32-48).

Therefore, based on the perspective of economic consequences of financial flexibility and Ortes-de-Mandojan and Bansal (2016), this paper adopts the standard deviation of the monthly stock return rate to measure the ability of firms to cope with environmental uncertainties, that is, financial flexibility. The smaller the standard deviation of the monthly stock return rate, the more stable the stock return; the smaller the financial volatility of firms and the stronger the ability to cope with exogenous shocks, the higher their financial flexibility (Oad Rajput et al.,2019). As a control, based on the perspective of the source of financial flexibility, we also use the sum of excess cash holdings and unused debt financing capacity to measure financial flexibility (lines 291-299, Table 3).

Point 2: As the authors acknowledge, endogeneity is a concern when interpreting the association between ESG performance and financial flexibility. I suggest the authors to clearly explain how the three tests in Section 4.3 help address the endogeneity concern in their setting. In addition, I suggest the authors to include firm fixed effects to further mitigate the influence of firm-level time-invariant factors.

Response 2: We carefully considered the reviewer's valuable comments and further explained how the three tests in Section 4.3 help address the endogeneity concern in their setting.

4.3.1. The independent variables lag by one period. Through this lead-lag method, the influence of ESG performance of the previous year on financial flexibility of the current year is explored, and the causal relationship between ESG performance and financial flexibility in time is tested, which alleviates the endogeneity problem to a certain extent (lines 420-423).

4.3.2. PSM regression analysis. The principle of PSM is to match an individual in the treatment group with at least one individual in the control group who is as similar as possible to the individual in other aspects, so as to eliminate the influence of inter-group differences to a certain extent and alleviate the endogeneity problem caused by observable variables (lines 427-430).

4.3.3. Heckman two-stage regression analysis. Heckman two-stage regression analysis can deal with the sample selection bias caused by unobserved data, and alleviate the research result bias caused by endogeneity to a certain extent. For this paper, firms with better ESG performance have more perfect management and information disclosure system, and we can obtain enough data for research. On the contrary, the lack of information disclosure system for firms with poor ESG performance makes it impossible to obtain sufficient research data. Therefore, we can only choose to abandon these samples, which leads to the problem of sample selection bias. In this paper, the Heckman two-step method is used for empirical test to alleviate the endogeneity problem caused by sample selection bias (lines 444-452).

We include both industry fixed effects and year fixed effects. The endogeneity test takes into account the influence of observable variables and unobservable data on the conclusion, which alleviates the influence of constant factors at the firm level to some extent. In addition, we also made a robustness check to ensure the reliability of the conclusion.

Point 3: I suggest the authors to discuss why examining their research question in the Chinese setting provides unique insights beyond studies that focus on the US or other countries.

Response 3: We have taken the valuable comments of the reviewers into full consideration. In the introduction section, we discuss why we study the relationship between ESG performance and financial flexibility in the context of China.

China is the world's second largest economy and an important emerging market. Since China's reform and opening-up, China's economic growth has made an increasing contribution to world economic growth. According to China's National Bureau of Statistics, China's economic growth contributed nearly 30 percent to world economic growth in 2018. China is the biggest contributor to world economic growth. Therefore, the study of China's economy has an important impact on world economic growth. As the micro subject of market, the sustainable development of firms is of great significance to economic growth. Due to the imperfect capital market in China and the widespread problem of information asymmetry, Chinese firms are faced with large financing constraints (Chan et al.,2012; Cull et al.,2015; Cheng et al.,2022). In the environment of increasing uncertainty, financing constraints lead to the obstruction of external financing of firms, unable to obtain enough funds to improve financial flexibility to cope with adverse impacts, and sustainable development faces serious challenges (lines 91-102).

Therefore, studying the relationship between ESG performance and financial flexibility in the Chinese context can provide unique insights.

We hope that our amendments will satisfy you. Once again, thank you for your efforts in revising and improving the quality of the manuscript.

Best Regards

         Dingzu Zhang

         Luqi Liu

2022.8.28

Reviewer 3 Report

Review of Sustainability-1889605

Does ESG Performance Enhance Financial Flexibility?  Evidence from China

             Your definition of financial flexibility on page 1 was the basis of the useful introduction to your paper.  Your explanation of the definition of the term creates a challenge.  Specifically, company profitability and retained earnings provide the basis for financial flexibility as you define it.  This definition would not be bad unless you restricted your operationalization of the term financial flexibility to measures of profit performance.

The material you present in the remainder of section 1 (through line 139) is well done, with one major exception.  You failed to show an appreciation for the components of ESG and their measurement.  By extension, you did not acknowledge that ESG standards and performance vary with the industry or industries being evaluated.  Thus, these are important considerations for the design of any research study on ESG topics.

Your paragraph on behavioral science perspectives (lines 159-169) states as conclusions several contentious issues.  Further, the relevance of the general topic to this paper’s investigation of financial flexibility in ESG is ignored.  The best solution may be to eliminate this paragraph from the paper.

The succeeding paragraph on the principal-agent perspective is similarly presented without rationale or apparent relevance.  ESG is the newest manifestation of decades-long concern for social responsibility broadly defined.  Your justification for basing the study on ESG hinges on your emphasis on its measurable performance characteristics.

The result of these underdeveloped and possibly secondary arguments is hypothesis 1, for which you have not provided a theory-based defense.

Your discussion of the social impact hypothesis (207-219) is interesting, but your lack of referencing raises serious questions about its development beyond the conceptualization stage.  If this line of thinking is critical to your study, the social impact discussion should be expanded and documented.

Hypothesis 2, which states that financial constraints play a mediating role in the relationship between ESG performance and financial flexibility, is not supported by an explanation for why the relationship should be expected, other than to remind us that financial constraints in part define strategic flexibility.  Again, you have presented a hypothesis that has not been shown to be supported conceptually or empirically by prior research.

A discussion of the WIND database (242) needs to be added to your paper.  Specifically, you need to explain the variables that comprised each ESG element, industry adjustments, calculated sub scores, and the analysis performed.  Until you do, I will be concerned about your interpretation of the results reported in Table 3.

The endogeneity migration analysis was valuable (345-356).

I appreciated the breadth and quality of your data analysis.  Unfortunately, the definitions of financial flexibility and financial performance lead to a near tautology, and the specification of industry-related ESG measures (and weighting) was underdeveloped.  In addition, the large number of variables led to the need for more interpretation than your hypotheses accommodated.  I note two examples:

1.      Your discussion of ESG-related results was hampered by your minimal attention to ESG measurement issues.  

2.      The analysis and interpretation of market disruption were limited by differences in industry contexts that were largely unexplored.

Your paper has promise.  Consider these changes:

·         Keep your focus on your title variables.  Delete the numerous side issues that distract a reader.

·         Strengthen your referencing.  Use the literature to reinforce your arguments.

·         Restructure your hypotheses.

·         Improve the operational definitions of your key variables.  Get specific about the composition of ESG – by industry.

Good luck as you continue your work on this project.

Author Response

Dear reviewer:

Thank you for your kind and good suggestions!

We have carefully revised it according to your guidance. In this version of the manuscript, we have mainly made the following adjustments:

Point 1: Your paragraph on behavioral science perspectives (lines 159-169) states as conclusions several contentious issues.  Further, the relevance of the general topic to this paper’s investigation of financial flexibility in ESG is ignored.  The best solution may be to eliminate this paragraph from the paper.

Response 1: Thanks for the reviewer's valuable comments, we eliminated behavioral science perspectives and added a signal theory perspective.

From the signaling perspective, good ESG performance often conveys a signal of sustainable development, which is an important basis for stakeholders to judge firm’s operational uncertainty and assess future profitability, cash flow and credit risk (Clarkson et al.,2008). ESG information can help alleviate information asymmetry, reduce risk expectations of stakeholders, improve credit availability of financial institutions, reduce debt financing costs, and en-hance debt financing ability (Eliwa et al.,2021). It also conveys a responsible and ethical signal to the outside world, which is conducive to establishing a good image of the firm. In this way, customers' subjective psychology can be positively improved, their evaluation and satisfaction with products can be enhanced, their purchase intention can be maintained (Hur et al.,2018; Chuah et al.,2020), and firm’s profitability can be improved. At the same time, it also strengthens the firm brand effect, increases the discrimination with similar firms and products, enhances the competition barrier, improves the profit space, thus increasing free cash flow (Oikonomou et al.,2014), which is conducive to enhancing cash flexibility (lines 182-194).

Point 2: Your discussion of the social impact hypothesis (207-219) is interesting, but your lack of referencing raises serious questions about its development beyond the conceptualization stage.  If this line of thinking is critical to your study, the social impact discussion should be expanded and documented.

Response 2: We carefully considered the reviewer's valuable comments, read more relevant literature, supplemented the literature related to social influence theory, and strengthened the citation of the article.

According to the social impact hypothesis, a firm’s behavior that damages social interests, such as environmental pollution, will lead stakeholders to doubt the firm’s performance ability and sustainable development ability (Shi et al.,2017). To safeguard their interests, the stakeholders who have the implicit claim on firm resources will transform the implicit contract into an explicit contract with higher cost for the firm by increasing the rate of return on investment and liquidated damages (Latané,1981; Cornell & Shapiro,1987; Hang et al.,2019), thus raising the financing cost of the firm and aggravating the financing constraints. On the contrary, firms with good ESG performance can attract and retain more high-quality employees, have higher production and operation efficiency, and have more substantial market competitiveness (Zhao et al.,2022) [63]. Therefore, stakeholders tend to believe that such firms are more powerful and moral, and their operational and default risks are relatively low (Chen et al.,2018; Pedersen et al.,2021), to proactively reduce expected risks and expected investment returns, which helps firms reduce financing costs, improve financing availability, and alleviate financing constraints (lines 243-255).

Point 3: A discussion of the WIND database (242) needs to be added to your paper.  Specifically, you need to explain the variables that comprised each ESG element, industry adjustments, calculated sub scores, and the analysis performed.  Until you do, I will be concerned about your interpretation of the results reported in Table 3.

Response 3: Thanks for the reviewer's valuable comments. In Section 3.2, we introduced the index composition of ESG rating system in detail. However, since the data provider (Shanghai Huazheng Index Information Service Co. Ltd.) did not disclose the proportion of each index and the specific evaluation method, we regret that we cannot have an in-depth discussion in this respect.

The index system has a total of nine grades, from C to AAA. Specifically, it consists of 3 first-level indicators, 14 second-level indicators and 26 third-level indicators. The first level indicators including three dimensions of environment, society and corporate governance. The secondary indicators of the environmental dimension include environmental management system, green business objectives, green products, external environmental certification and environmental violations. The three-level indicators of the environmental dimension include environmental management system, low carbon plan or target, green business plan, carbon footprint, sustainable products or services, products or firms obtaining environmental certification, and environmental violations. The secondary indicators of the social dimension are institutional system, health and safety, social contribution and quality management. The three-level indicators of the social dimension are the quality of social responsibility reporting, the goal or plan to reduce safety accidents, negative business events, business accident occurrence trend, social responsibility related donations, employee growth rate, rural revitalization, and product or firm obtaining quality certification. The secondary indicators of corporate governance dimension include system construction, governance structure, business activities, business risks and external disposition. The three-level indicators of corporate governance dimension include corporate self-supervision, affiliated transactions, board independence, tax transparency, asset quality, overall financial credibility, short-term debt repayment risk, equity pledge risk, information disclosure quality, violation events of listed companies and subsidiaries, and violation events of executives and shareholders. For a clearer rating system, please contact Huazheng (https://www.chindices.com/esg-ratings.html#esg-ratings-methodology) (lines 308-330).

Point 4: Hypothesis 2, which states that financial constraints play a mediating role in the relationship between ESG performance and financial flexibility, is not supported by an explanation for why the relationship should be expected, other than to remind us that financial constraints in part define strategic flexibility.  Again, you have presented a hypothesis that has not been shown to be supported conceptually or empirically by prior research.

Response 4: We carefully considered the reviewer's valuable comments, and for hypothesis 2, we read more literature and explained why the mediating effect of financing constraints should be expected.

Financing constraints refer to the difficulty of raising funds relative to investment opportunities, which is rooted in information asymmetry in the incomplete market (Deng & Zeng,2014; Banerjee et al., 2020). Studies have found that financing constraints affect business performance (Pathan et al.,2016), reduce total factor productivity (Krishnan et al.,2015), restrict outbound investment (Buch et al.,2014), and inhibit R&D and innovation (Zhao et al.,2019), thus damaging the sustainable development ability of firms. With the deepening of environmental uncertainty, Chinese firms are increasingly faced with financing constraints. Financing constraints restrict firms' access to external funds, impair financing ability, and thus are not conducive to improving financial flexibility (Laghari et al.,2022). It also inhibits the growth of firms and reduces their risk coping ability (Moscalu et al.,2020; Peng et al.,2019) [55,56]. It is of great significance for firms to explore how to alleviate financing constraints to im-prove financial flexibility, enhance risk coping ability and realize sustainable development under uncertain environment (lines 223-234). 

Point 5: Keep your focus on your title variables.  Delete the numerous side issues that distract a reader. Restructure your hypotheses. Strengthen your referencing. 

Response 5: Thanks for the reviewer's valuable comments. In Section 2, We re-sorted the relevant literature, improved our theoretical analysis, strengthen our referencing and use the literature to reinforce our arguments. We also delete the numerous side issues that distract readers to make our paper more reasonable (see Section 2).

We hope that our amendments will satisfy you. Once again, thank you for your efforts in revising and improving the quality of the manuscript.

Best Regards

         Dingzu Zhang

         Luqi Liu

2022.8.28

Round 2

Reviewer 2 Report

The authors have done a great job responding to my comments in the last round. My only suggestion is to use the alternative measure of financial flexibility (FF_C&D) in all subsequent tests beyond Table 3. Also, this alternative variable is not a control (line 299).

Author Response

Dear reviewer:

    Thank you for your kind and good suggestions!

     We have carefully revised it according to your guidance. In this version of the manuscript, we have mainly made the following adjustments:

    Point 1: My only suggestion is to use the alternative measure of financial flexibility(FF_C&D)in all subsequent tests beyond table 3. Also, this alternative variable is not a control.

    Response 1: Thanks to the reviewer's valuable comments, we conducted an empirical analysis using financial flexibility (FF_C&D) in all subsequent tests beyond table 3. As shown in Table 4, Table 5, Table 6 and Table 7.

    We hope that our amendments will satisfy you. Once again, thank you for your efforts in revising and improving the quality of the manuscript.

   Best Regards

         Dingzu Zhang

         Luqi Liu

2022.9.4

Reviewer 3 Report

You made several useful adjustments to the manuscript but left many issues unaddressed. The hypotheses still need to be rewritten. The operationalization of key variables is sometimes vague (e.g., ESG, financial flexibility), and the limitations of using a dataset not collected for your study should be acknowledged explicitly. 

The literature review relies on mixed-quality sources. The paper would be stronger if you reduced the bibliography by 40-50% by keeping only the highly rated journal articles.

The results will be stronger if you limit your discussion to your hypotheses and the consequences of your findings on theory building.

Good luck as you continue your work on this project.

Author Response

Dear reviewer:

Thank you for your kind and good suggestions!

We have carefully revised it according to your guidance. In this version of the manuscript, we have mainly made the following adjustments:

Point 1: The hypotheses still need to be rewritten. The operationalization of key variables is sometimes vague (e.g., ESG, financial flexibility).

Response 1: Thanks for your valuable comments, we've rewritten our hypotheses and more clearly illustrated the operationalization of the key variable (financial flexibility, ESG).

Financial flexibility (FF): Based on the perspective of economic consequences of financial flexibility and Ortiz-de-Mandojan and Bansal [52], this paper adopts the standard deviation of the monthly stock return rate to measure the ability of firms to cope with environmental uncertainties, that is, financial flexibility (FF_STO). Firstly, we download the monthly stock returns of Chinese firms from 2015 to 2020 from RESSET database. Secondly, EXCEL is used to collate and analyze the data, and samples with missing key data were deleted. Finally, STDEVP function is used to calculate the standard deviation of monthly stock returns in each year. The smaller the standard deviation of the monthly stock return rate, the smaller the financial volatility of firms, the stronger the ability to cope with exogenous shocks, and therefore the higher their financial flexibility. To reduce the bias of the data measurement and based on the perspective of the source of financial flexibility, we also use the sum of excess cash holdings and unused debt financing capacity to measure financial flexibility (FF_C&D). First, we download the cash ratio and asset-liability ratio of Chinese firms from 2015 to 2020 from the CSMAR database. Secondly, EXCEL is used to calculate the industry average cash holding ratio and industry average debt ratio. Finally, we calculate the financial flexibility (FF_C&D) according to the following formula:

FF_C&D = (cash holding ratio of firm - average cash holding ratio of industry) + MAX (0, average debt ratio of industry - debt ratio of firm)

ESG performance (ESG): Referring to the research of Wang et al. [53], ESG rating data disclosed by Shanghai Huazheng Index Information Service Co. Ltd. is utilized. We download the quarterly ESG rating data of Chinese firms from the WIND database from 2015 to 2020, the 9 grades of this index "C, CC, CCC, B, BB, BBB, A, AA, AAA" were assigned 1-9 respectively, and the natural logarithm was used to measure the quarterly ESG performance. Then, we use quarterly ESG rating data to calculate annual average ESG performance as a measure of corporate ESG performance (lines 286-312).

Point 2: The limitations of using a dataset not collected for your study should be acknowledged explicitly and the paper would be stronger if you reduced the bibliography by 40-50% by keeping only the highly rated journal articles.

Response 2: We carefully considered your valuable comments, more explicitly acknowledged the limitations of the dataset we used. We have reduced some of the insignificant bibliographies and kept only the highly rated journal articles.

First, we only use firms from mainland China as samples to explore the relationship between ESG performance and financial flexibility, so an analysis based on international data is lacking. Second, in terms of the measurement of the dependent variable, we use the methods commonly used in the literature -- the standard deviation of monthly stock return and the sum of cash flexibility and debt flexibility to measure financial flexibility indirectly. We fail to use creative methods to measure financial flexibility directly, and cannot provide suggestions for how much financial flexibility to reserve. Third, considering that we only use Chinese firms as samples, we also only use ESG rating data disclosed by Huazheng which is more consistent with China's national conditions, failing to compare the impact of ESG rating data from different institutions on conclusions, and failing to provide suggestions for firms on how to invest in ESG. Fourth, in addition to analyzing all samples as a whole, we only consider the differences in the relationship between ESG performance and the financial flexibility of manufacturing firms under different environmental uncertainties and market concerns without con-ducting in-depth research on more industries (lines 724-743).

Point 3: The results will be stronger if you limit your discussion to your hypotheses and the consequences of your findings on theory building.

Response 3: Thanks for your valuable comments. In Section 6, we refined the content of the discussion.

At present, the degree of environmental uncertainty is deepening, and the sustainable development of firms is facing severe challenges. In the uncertain environment, firms need to reserve appropriate financial flexibility to cope with risks, stable operation, seize potential investment opportunities, and achieve sustainable development. ESG performance provides additional information to stakeholders and is an important con-sideration for stakeholders in their investment decisions. Good ESG performance can help firms gain the trust and support of stakeholders, thereby enhancing the cash holding and external financing ability.

The results in Table 3 support H1, good ESG performance is beneficial to improve financial flexibility. Our results support the views of Hur et al. (2018), Olsen (2017) and Ng & Rezaee (2015), providing evidence of instrumental stakeholder theory and signaling theory. Firms can obtain scarce resources for sustainable development by managing stakeholder relationships. For example, firms with good relations with the government can convey political advantages to the market, gain recognition and trust from creditors and investors, and thus have more financing convenience, lower financing cost and stronger financing ability. Good ESG performance can convey a positive signal of responsibility and ethics to the outside world, strengthen brand effect, enhance customer satisfaction and purchase intention of products, and improve corporate profitability and cash flow level. The improvement of financing ability, profitability and cash flow makes firms have more sufficient resources to cope with adverse shocks, so as to improve the ability of firms to cope with environmental uncertainties and enhance financial flexibility.

Information asymmetry is the root cause of financing constraints. Due to the imperfect capital market in China, Chinese firms are generally faced with financing constraints. Under the increasingly severe environmental uncertainty, how to alleviate financing constraints, enhance financing ability, improve financial flexibility, and avoid falling into financial distress is an urgent problem for firms to achieve sustainable development. The results in Table 3 also support H2, financing constraints have a mediating role in the process of ESG performance affecting financial flexibility. Our results support Deng et al. (2013) and Latane (1981) and provide evidence for social influence theory. Behaviors that violate the ESG concept will make stakeholders doubt the sustainable development ability of the firm, thereby increasing the return required by investors, raising the financing cost of the firm, aggravating financing constraints, and thus hindering the improvement of financial flexibility. On the contrary, improving ESG performance of firms can help improve organizational reputation, reduce information asymmetry, thus enhance investor confidence, alleviate financing constraints, raise funds, and improve financial flexibility. Therefore, ESG performance improves financial flexibility by releasing financing constraints.

Environmental uncertainty increases the risk of firm operation and default, deteriorates the financing environment and operation environment of firms, and poses a great threat to sustainable development. Then, does the degree of environmental un-certainty affect the relationship between ESG performance and financial flexibility? The results in Table 6 suggest that ESG performance is more conducive to improving financial flexibility when the degree of environmental uncertainty is high. However, this enhancement effect is only reflected in improving the risk coping ability of firms, while hindering firms to hold cash and reserve surplus debt capacity. This may be because ESG performance acts as an insurance against the negative effects of an uncertain environment (LINS et al.,2017). But firms may take on more debt in response to external shocks, and investing in ESG further reduces the firm's cash, thereby compromising its ability to hold cash and spare debt.

Market attention has loudspeaker and authentication functions, which can improve the speed and authority of ESG information transmission, thus enhancing the market response of ESG information. As a limited resource, the degree of market attention may have an impact on the relationship between ESG performance and financial flexibility. The results in Table 7 suggest that the impact of ESG performance on financial flexibility is stronger when market concern is high. With the increase of market attention, the transmission efficiency and persuasion of ESG performance in the capital market are improved, which is beneficial to strengthen the release effect of ESG performance on financing constraints and better improve financial flexibility (Barber & Odean,2008).

We hope that our amendments will satisfy you. Once again, thank you for your efforts in revising and improving the quality of the manuscript.

Best Regards

         Dingzu Zhang

         Luqi Liu

2022.9.4
